# An Architecture Search Framework for Inference-Time Techniques

**Jon Saad-Falcon** [1]  **Adrian Gamarra Lafuente** [1]  **Shlok Natarajan** [1]  **Nahum Maru** [1]  **Hristo Todorov** [1]
**Etash Guha** [2]  **E. Kelly Buchanan** [1]  **Mayee Chen** [1]  **Neel Guha** [1]  **Christopher Ré** [1]  **Azalia Mirhoseini** [1]

## Abstract

Inference-time techniques, such as repeated sampling or iterative revisions, are emerging as powerful ways to enhance large-language models (LLMs) at test time. However, best practices for developing systems that combine these techniques remain underdeveloped due to our limited understanding of the utility of each technique across models and tasks, the interactions between them, and the massive search space for combining them. To address these challenges, we introduce ARCHON, a modular and automated framework for optimizing the process of selecting and combining inference-time techniques and LLMs. Given a compute budget and a set of available LLMs, ARCHON explores a large design space to discover optimized configurations tailored to target benchmarks. It can design custom or general-purpose architectures that advance the Pareto frontier of accuracy vs. maximum token budget compared to top-performing baselines. Across instruction-following, reasoning, and coding tasks, we show that ARCHON can leverage additional inference compute budget to design systems that outperform frontier models such as OpenAI's o1, GPT-4o, and Claude 3.5 Sonnet by an average of 15.1%.

## 1. Introduction

Inference-time techniques—strategies that use additional compute during model inference—are gaining traction as effective methods for improving model capabilities. LLMs, such as OpenAI's o1 (OpenAI, 2024), QwQ (Team, 2024), and Sky-T1 (Team, 2025), utilize such techniques to translate additional inference compute into better performance across a broad set of tasks. Example techniques include generation ensembling, ranking, and fusion, where models in the

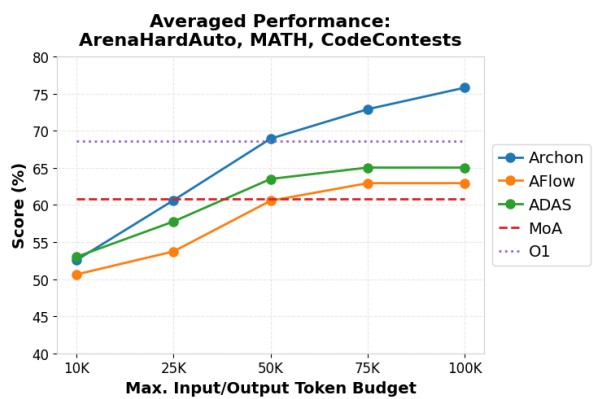

Figure 1: **ARCHON's Performance Effectively Scales with Increasing Inference Budget.** Individual dataset analysis included in Figure 8.

ensemble are queried in parallel, their responses are ranked, and the best ones are fused into a single, higher quality output, respectively (Jiang et al., 2023b; Wang et al., 2024a). Other types of inference-time techniques are based on querying a single LLM successively (via repeated sampling) and using a voting strategy or unit tests to select the top generation (Brown et al., 2024; Chen et al., 2024; Li et al., 2024a).

Recent work has made progress towards building robust *inference-time architectures*: systems composed of one or more large language models (LLMs) leveraging inference-time techniques. Examples include Mixture-of-Agents (MoA) (Wang et al., 2024a) and LLM-Blender (Jiang et al., 2023b), as well as single-model systems like ADAS (Hu et al., 2024) and AFlow (Zhang et al., 2024). However, our experiments show that these top-performing baselines have limitations in compute utilization and task generalization. (see Section 4.2). We argue that designing effective and generalizable inference-time architectures requires the following:

- **Understanding the Utilities of Inference-Time Techniques**: Inference-time architectures typically delegate their additional inference budget towards more model sampling calls (Chen et al., 2024; Brown et al., 2024), which can be effective for math and coding tasks. Other tasks, such as following instructions and reasoning, have been shown to benefit from additional techniques,

[1]Stanford University, Stanford, CA, USA [2]University of Washington, Seattle, WA, USA. Correspondence to: Jon Saad-Falcon <jonsaadfalcon@stanford.edu>.

*Proceedings of the 42nd International Conference on Machine Learning*, Vancouver, Canada. PMLR 267, 2025. Copyright 2025 by the author(s).

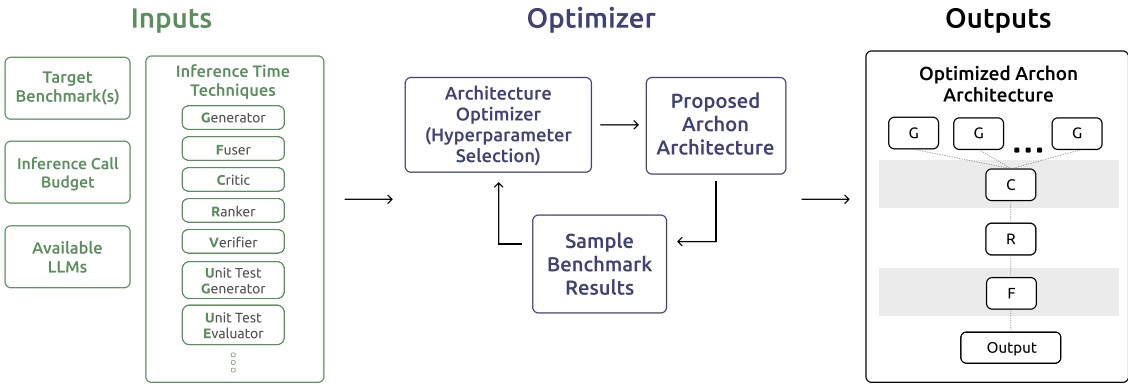

Figure 2: **Overview of ARCHON Framework**: ARCHON's search algorithm requires the following inputs: target benchmarks, inference call budget, available LLMs, and available inference-time techniques (**left**). The search algorithm uses Bayesian optimization (Snoek et al., 2012) to construct and evaluate different ARCHON configurations (**middle**) before returning the optimized ARCHON architecture (**right**) for the target benchmarks (Section 3.3).

including ranking and fusion (Wang et al., 2024a; Jiang et al., 2023b). While all of these methods are valuable, *it is essential to identify which inference-time techniques are most effective for different task categories.*

- **Understanding the Interactions Between Inference-Time Techniques**: While previous studies analyzed these techniques individually (e.g., generation sampling in Chen et al. (2024)), *we need a more comprehensive understanding of the relationships between different inference-time techniques* across different tasks (e.g., is it better to use more models or generate more samples per model?).
- **Efficiently and Automatically Searching the Large Design Space of Inference-Time Architectures**: Given a set of available LLMs and target tasks, there is currently no single prevailing inference-time architecture for maximizing downstream accuracy across all tasks (Table 1). The search space for inference-time architectures is expansive, requiring practitioners to make several key configuration decisions, such as *which LLMs to use, how many times to sample them, and how to combine and filter the candidate generations*. These motivate the need for automated and adaptive architecture search approaches.

In our work, we address each of these challenges. First, we **evaluate the utilities of a comprehensive set of existing and proposed inference-time techniques** across instruction-following, reasoning, and coding tasks. Using both open-source and closed-source models, we examine a range of techniques such as *ensembling, fusion, ranking, critiquing, verification, and model-based unit test generation/evaluation* (Sections 3.1 and 3.2). We find that no single technique completely dominates across all tasks, with different approaches being more effective for different tasks.

Second, we **analyze the interactions between inference-time techniques** and explore the benefits of adding new models and new techniques individually. We find that generation ensembling combined with critique, verification, and fusion improves the final response quality beyond the oracle best candidate from individual (non-fused) responses, particularly for instruction-following and reasoning tasks (Figure 4; Figure 7; Table 5). We also demonstrate increased performance as we scale up the layers of inference-time techniques and combine multiple approaches together, allowing us to discover effective new combinations of inference-time techniques (Sections 3.2, 4.2, A.3). Combining multiple strategies significantly improves task performance, but determining the specific combination remains challenging. This requires manually testing models, inference-time techniques, architecture designs, inference budgets, and more.

Third, drawing upon our analysis of inference-time techniques, we present **ARCHON**, an open-source modular framework for automatically designing LLM systems composed of existing inference-time techniques (or new ones), allowing practitioners to optimize for their desired objective functions: accuracy, latency, and cost (Sections 3.1, 3.3). Unlike alternative LM systems that perform prompt engineering and tool use over a single LM (Khattab et al., 2023; Yuksekgonul et al., 2024; Hu et al., 2024; Zhang et al., 2024), our approach integrates multiple LMs in a single architecture and reduces prompt selection to a set of core components. The ARCHON framework utilizes automatic architecture search algorithms to maximize generation quality for the given tasks(s), leveraging Bayesian optimization (Snoek et al., 2012; Nardi et al., 2019) techniques inspired by (NAS) (Zoph & Le, 2017; Ren et al., 2021) to rapidly traverse the space of potential inference architectures (Section 3.3).

We evaluate ARCHON architectures across a diverse set

of instruction-following, reasoning, and coding benchmarks (Table 1): MT-Bench, Arena-Hard-Auto, Alpaca-2.0 Eval, MixEval, MATH, and CodeContests (Zheng et al., 2023; Li et al., 2024b; 2023; Ni et al., 2024; Hendrycks et al., 2021; Li et al., 2022). Our best ARCHON architectures surpass both frontier models (e.g. OpenAI's O1, GPT-4o and Claude-3.5 Sonnet) and prior top-performing inference-time architectures (e.g. ADAS, AFlow, and MoA), *boosting state-of-the-art (SOTA) performance by 15.1%*, on average. Furthermore, ARCHON achieves SOTA performances while using *20.0% less inference calls, 15.1% less input tokens, and 13.5% less output tokens* than alternative inference-time architectures (Figure 1; Table 1; Figure 8). Even when solely using open-source LLMs, ARCHON architectures, on average, surpass SOTA LLMs by 11.2%.

Overall, we present ARCHON as an open-source inference-time framework, readily extensible to new inference-time techniques, models, and tasks via user-friendly interfaces.

## 2. Related Work

Despite advancements in inference-time architectures, many architectures focus on additional generations (Jiang et al., 2023b; Chen et al., 2024; Davis et al., 2024), which is effective for reasoning tasks (Brown et al., 2024). However, for tasks like instruction-following and reasoning, techniques such as fusion and ranking are effective for bolstering task performances (Wang et al., 2024a; Jiang et al., 2023b). Prior studies have explored limited aspects of configurations, often focusing on specific benchmarks (Jiang et al., 2023b; Wang et al., 2024a; Chen et al., 2024; Li et al., 2024a). It's crucial to efficiently develop inference-time architectures, as optimal configurations vary based on benchmarks, available models, and inference compute limits (Section 4.2). Furthermore, LM orchestration frameworks, such as DSPy (Khattab et al., 2023), only optimize a single prompt for a single LM, better equipping it for tool use by utilizing supervised data but still unable to leverage multiple inference-time techniques in parallel or sequentially. While each of these approaches manually selects a subset of existing techniques, ARCHON unifies available inference-time techniques and automates architecture construction with search algorithms, simplifying the model and component selection process for each set of tasks (Sections 3.1 and 3.3).

## 3. Inference-Time Techniques for ARCHON

With the proliferation of inference-time techniques, ARCHON introduces a systematic framework for understanding and unifying these methods into inference-time architectures. Below, we elaborate on the structure, inputs, and outputs of each of the inference-time techniques (Table 3). Then, we discuss how to combine the different techniques into an inference-time architecture (Section 3.2) before finally exploring automatic approaches for constructing inference-time architectures (Section 3.3).

### 3.1. LLM Components of ARCHON

In this section, we discuss the *LLM components* of ARCHON, which are LLMs that perform a specific inference-time technique. We test an array of different components inspired by recent work, incorporating approaches for generating, ranking, and fusing candidates (Wang et al., 2024a; Jiang et al., 2023b) as well as approaches for improving candidate response quality through critiquing, verifying, and unit testing (Bai et al., 2022; Zheng et al., 2023). The components and their prompts are summarized in Table 3 and Appendix A.2. We also perform an extensive ablation study of the given ARCHON components across instruction-following, reasoning, and coding benchmarks to better understand their individual utilities and their optimal combinations for different tasks (Appendix A.3).

**Generator** is an LLM that takes in the instruction prompt and outputs candidate responses. Generators can be called in parallel to perform *generation ensembling* (i.e. calling multiple LLMs in parallel) (Wang et al., 2024a), or sampled multiple times (Brown et al., 2024). The number of models, samples, and generation temperature can be adjusted.

We find additional model sampling to significantly boost performance (Figure 6), particularly for coding tasks (Table 1). We see a similar pattern for model ensembling, where sampling from additional models leads to continual performance increases (assuming the models are ordered from best to worst for the given task) (Figure 7).

**Fuser** is an LLM that, given an instruction prompt and a set of proposed responses as input, combines these responses to generate one or more higher-quality fused responses.

For every benchmark explored, we found that the Fuser module substantially improved performance (8.9% on average) (Figure 6; Figure 7; Figure 4). Additionally, we observed similar benefits in the ARCHON framework when adding multiple layers of Fusers (Figure 4). The number of Fuser layers needed to improve performance varied by task (Figure 12), with some tasks receiving limited benefits from added layers (1-2 point increase in accuracy for MixEval) while others experienced significant benefits with 3-4 fusion layers and more (10 to 15 point increase in win rate for MT Bench and Alpaca Eval 2.0).

**Ranker** is an LLM that, given an instruction prompt and a set of proposed responses as input, ranks the candidate generations based on their quality, producing a ranked list of responses as output. This ranking is then used to filter the set of responses to the top-$K$, as specified.

From our results in Table 5, Figure 6, and Figure 7, our results show the Ranker was most effective for instruction-following and reasoning tasks by using pair-wise comparisons that

focus on style and prompt adherence. We found that on MT Bench and Arena-Hard-Auto benchmarks, the Ranker improved output quality by 10.8% over random selection while performing within 2.7% of oracle selection.

**Critic** is an LLM that, given an instruction prompt and a set of proposed responses as input, produces a list of strengths and weaknesses for each response, which is then used to improve the quality of the final response (Section 3.2; Figure 4).

The Critic module proved effective for every task we explored in Figure 4 and Table 5. With our 10-model 70B+ Generator ensemble and Fuser configuration of ARCHON, the added Critic improved performance on average by 11.5 percentage points across the benchmarks explored.

**Verifier** is an LLM that verifies whether a provided candidate response has appropriate reasoning for a given instruction prompt. It proceeds in two stages: **Stage #1** takes in the instruction prompt and a candidate response as input and outputs reasoning for why the candidate response is correct; **Stage #2** takes in the instruction prompt, candidate response, and produced reasoning before outputting reasoning and a verdict (i.e., binary [Correct] or [Incorrect]) for whether or not the candidate response is correct according to the provided instruction prompt and reasoning. Only verified responses are passed to the next ARCHON layer.

The Verifier was most effective for the reasoning benchmarks explored in Table 5, improving performance by 8.4% for MixEval, MixEval Hard, and MATH. When just using a 70B+ Generator ensemble with Verifier module after generation, the ARCHON configuration lagged behind the ARCHON ensemble and fuser configuration by 1.5%, on average, across all benchmarks explored, suggesting verification is most effective when combined with other inference-time techniques.

**Unit Test Generator** and **Unit Test Evaluator** are complementary LLM components in our system: the Unit Test Generator takes an instruction prompt and produces 5-10 concise test statements (Section 4.2; examples in Table 16) for assessing response accuracy and relevance, while the Evaluator takes the instruction prompt, candidate response(s), and these tests as input to rank responses by test passage. The Evaluator justifies and aggregates test verdicts across candidates, scoring each response for reasoning and coding tasks, and only responses passing all tests proceed to the next ARCHON layer. This approach extends evaluation beyond coding to various task types through configurable test quantities.

The Unit Test Generator and Evaluator were most effective on reasoning and coding tasks, improving performance on benchmarks that required more verification steps (7.4% boost) (Table 5). When the 70B+ ensemble of Generators was only combined with unit tests, it was less effective for reasoning tasks like Arena-Hard-Auto and MixEval, lagging behind the ensemble and fuser configuration by

3.1%. However, when we increased generation sampling and added unit test generation/evaluation for CodeContests, we observed a 56% boost in Pass@1 performance (Table 1), increasing from 17.9 to 29.3% Pass@1.

## 3.2. Combining the LLM Components

**Performance Gains from Scaling Inference-Time Techniques**: We explore the utilities of individual AR-CHON components and evaluate whether combinations of inference-time techniques enable us to *build LM systems greater than the sum of their parts*. For our analysis, we look at seven datasets spanning instruction-following, reasoning, mathematics, and coding: MT-Bench (Zheng et al., 2023), AlpacaEval 2.0 (Li et al., 2023), Arena Hard Auto (Li et al., 2024b), MixEval (Ni et al., 2024), MixEval-Hard, MATH (Hendrycks et al., 2021), and CodeContests (Li et al., 2022). We also test across the current SOTA open-source and closed-source LMs (Table 28; Table 29). For the analysis of each inference-time technique, we focus on **1)** testing it across different benchmarks, **2)** scaling its usage individually, **3)** scaling it while randomly choosing another technique and holding that technique constant, and **4)** varying its position among different components. We include these ablation experiments in Section A.3, where we include the ARCHON component combinations in Table 5 and the model type used in the combinations in Table 6 and Table 9.

From our analysis, we find several trends (designated with **T**s) across the combinations of inference-time architectures:

- **T1**: Repeated model sampling and additional ensemble models leads to substantial gains, leading to 9.3% and 18.5% increases, respectively (Figure 11; Figure 7).
- **T2**: Scaling the layers of inference-time techniques significantly improves performance across instruction-following, reasoning, and coding tasks, such as always adding a single fuser as the last layer (Figure 4).
- **T3**: Scaling the diversity of inference-time techniques included also bolsters task performance across the explored tasks, with critics and rankers before fusers being particularly effective (Figure 4; Figure 11).
- **T4**: In reasoning tasks, incorporating the Verifier and Unit Test Generator/Evaluator modules alongside the Fuser improves performance by filtering out flawed responses, contributing to significant performance gains in tasks like MixEval and CodeContests (Table 5; Section A.9).

**Framework Overview**: Drawing upon our analysis of the inference-time components, we propose **ARCHON**, a framework for automatically designing LLM systems composed of existing inference-time techniques (or new ones). Inspired by the structure of neural networks (Hinton et al., 1992), ARCHON consists of layers of LLM components (Figure 2; Section 3.1). Each layer is composed of sets of LLM components called in parallel. These components perform a

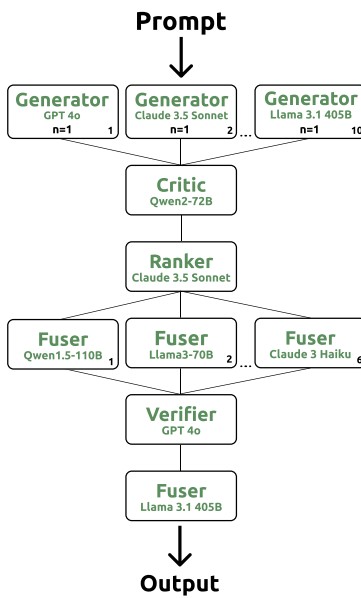

**Prompt**

Figure 3: **Example ARCHON Architecture**: This architecture starts with ten generator models (each sampled once), followed by a critic model, a ranker model, one layer of six fuser models, a verifier model, and finishes with a fuser model.

text-to-text operation on the initial instruction prompt and the candidate responses from the previous layer. Furthermore, like a neural network, some layers perform *transformations* of the provided list of strings (e.g., Generator and Fuser), converting a list of strings into a different list of strings (the numbers of candidates can vary from the original number of candidates). Other components introduce non-linearities into the ARCHON structure, performing filtering of the list of strings (e.g., Ranker and Verifier). Ultimately, the inputs and outputs for each layer is always a list of strings, whether that is the instruction prompt (i.e., a single string) or a list of candidate responses. If a list of strings is outputted at the last layer of the ARCHON structure, the first string in the list is returned.

Unlike a classical neural network, no weights are learned between the LLM components and the layers; in turn, the ARCHON architecture can be deployed off-the-shelf without any tuning. Additionally, a single state is transformed sequentially from the input layer to the final output; this single state is the initial instruction prompt and the current candidate responses (example architecture in Figure 3).

**Rules for Construction**: The LLM components in Section 3.1 can only be placed in specific orders (4). While alternative combinations and orderings of ARCHON components are technically viable, we found these orderings to be optimal after conducting an ablation study of ARCHON components across seven benchmarks and two model classes (open-source and closed-source) (Appendix A.3).

1. Only one type of component is allowed in any given layer.
2. Generator components can only be placed in the first layer of ARCHON; you can place one or more Generators.
3. The Critic must come before a Ranker or a Fuser. Otherwise, the generated strengths and weaknesses cannot be incorporated into generation ranking or fusion.
4. Ranker, Critic, Verifier, and Unit Test Generator/Evaluator layers can go anywhere in ARCHON except the first layer. For each of these components, it must be the only module in its layer.
5. Fuser components can go anywhere in ARCHON except the first layer. Multiple Fusers can be used in a layer.
6. Unit Test Generators and Evaluators are placed in consecutive layers, with the Unit Test Generator always first.

### 3.3. Architecture Search Algorithms

**Search Hyperparameters**: In this section, we explore how to automatically design inference-time architectures for target tasks via ARCHON's architecture search algorithms. Guided by the trends found in our analysis in Section 3.2, we establish six axes of hyperparameters for the search space:

1. **Top-$K$ Generators for Ensemble**: The top-$K$ models for the initial Generator ensemble, ranging from 1 to 10 (**T1**). The top-$K$ models are selected greedily based on their individual performances on target task(s)(Table 29).
2. **Top-$K$ Generator Samples**: The number of samples gathered from each ensemble generator (same for all the models), ranging from 1 to 5 (**T1**). For CodeContests, we explore high-sample settings: [1, 10, 100, 500, 1000].
3. **Number of Fusion Layers**: Ranges from 1 to 4. The last fusion layer will always have a single Fuser (**T2**).
4. **Top-$K$ Fusers**: Number of models used for each fusion layer, ranges from 2 to 10 in increments of 2 (**T2,3**).
5. **Critic and Ranker Layers**: We add critic and ranker layers before each fuser layer since we find they provide added benefits across the benchmarks explored (**T3**) (Section 3.2; Figure 4; Figure 7).
6. **Evaluation Layer**: Option to add Verifier, Unit Test Gen./Eval., or neither before the last Fuser layer (**T4**).

While it is possible to further expand the search space of potential ARCHON architectures (e.g., different temperatures for generative LLM components, alternative prompts for each LLM component, additional LLM components for ARCHON, etc.), the trends we identify from Section 3.2 reasonably constrain the search space of configurations to focus on the most influential hyperparameters. In total, our search space contains 9,576 configurations, which we obtain by combining all possible hyperparameters and removing invalid configurations (for example, we discard configurations where the number of initial generations exceeds the context window of the fusers).

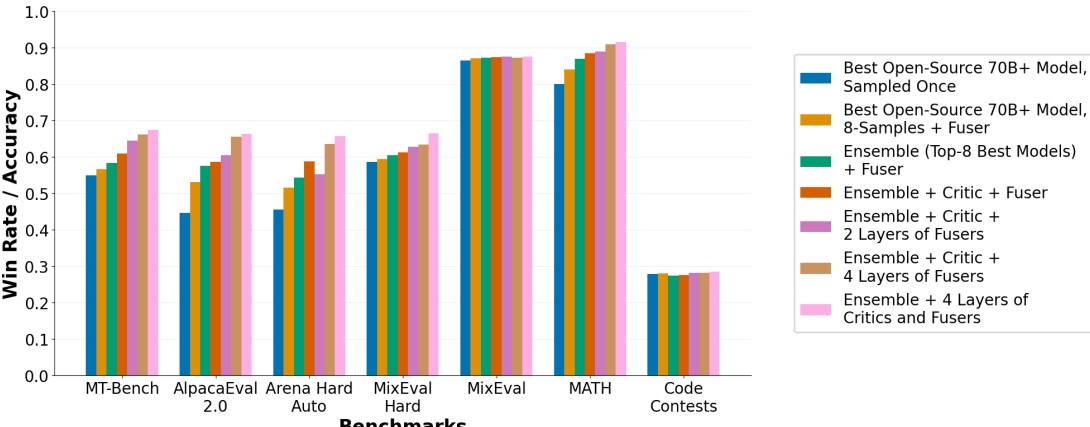

Figure 4: **Performance Improves by Scaling *Layers* of Inference-Time Techniques**: When controlling for inference budget, generation ensembling and fusion across 8 different 70B LLMs is generally more effective than repeated sampling with only the top performing model. Furthermore, adding layers of critique and fusion led to a 18.8% boost in task performance, on average. However, the best inference-time architecture differed by task, such as MixEval and CodeContests (Section 4.3), which inspired us to develop architecture search techniques for ARCHON (Section 3.3).

**Search Method**: The ARCHON search method takes in four inputs: the target benchmark(s), the inference call budget, the set of available LLMs, and the inference-time techniques for construction (Figure 2). As output, the search method outputs a single optimized ARCHON architecture. We use 20% of each target dataset as a development set for guiding architecture search. We explore three approaches for ARCHON's architecture search: *random search* (randomly test potential architectures in the search space), *greedy search* (greedily optimize individual hyperparameters one at a time, starting from a random initial architecture), and *Bayesian Optimization* (Snoek et al., 2012) (global hyperparameter optimization with Gaussian processes). As inputs, Bayesian optimization takes in a vector specifying the configuration choices for the generators (i.e., number of models and samples), layers of fusers, numbers of fusers per layer, and final verifier / unit tester (Section 3.2). Bayesian optimization begins by sampling a specified number of random ARCHON architectures to calibrate its surrogate model. The task performance of these sampled architectures is used to guide more informed architecture suggestions during the configuration search. The algorithm repeats the following cycle—evaluating each suggested architecture and using its performance to refine future suggestions—until it discovers the optimal ARCHON configuration, or until the inference call budget is exhausted. For more details on our open-source Bayesian optimization approach, please see Appendix A.4, where we further discuss implementation and how to utilize alternative optimization functions, such as latency.

Bayesian optimization found the best architectures in 96.0% of searches and required 88.5% fewer architecture evaluations than greedy search and 90.4% fewer than random search (Figure 10). The effectiveness of Bayesian optimization increases with the number of initial randomly sampled architectures, up to around 230-240 samples, after which further testing is better focused on configuration search (Table 20). For limited inference call budgets (<20 calls), Bayesian optimization is less effective, and traditional methods like greedy search may perform comparably (Table 21).

**Adding Search Restrictions**: To impose compute constraints during architecture search, we exclude any ARCHON architecture that would exceed the inference call, input token, or output token budgets from the search space. Multiple restrictions can be added. For example, you can filter out architectures with more than 20 inference calls or more than 20,000 input tokens. This prevents our Bayesian optimization algorithm from even considering these invalid architectures in our architecture search, allowing us to compute-match ARCHON against alternate inference-time frameworks such as ADAS and AFlow (Figure 8; Figure 5).

## 4. Experiments

Our experiments focus on answering the following questions: **(1)** how does ARCHON compare to existing SOTA LLMs and inference-time architectures in terms of accuracy and compute efficiency (Section 4.2)? **(2)** how does ARCHON performance compare across the tasks explored (Section 4.3)? **(3)** what are the considerations for model size, latency, and cost surrounding ARCHON (Section 4.4)? We outline the benchmarks, models, and techniques for constructing ARCHON architectures in Section 4.1.

| | | Approaches | Average Infer. Calls | Average Input Tokens | Average Output Tokens | Avg. PFLOPs per Query | Dollars per Query | MT Bench W.R. | Alpaca Eval 2.0 L.C. W.R. | Arena Hard Auto W.R | MixEval Hard Acc. | MixEval Acc. | MATH* Pass @1 | Code Contests* Pass @1 |
|---|---|---|---|---|---|---|---|---|---|---|---|---|---|---|
| **Baselines** | **LM** | GPT-4o | 1 | 95 | 549 | 0.6 ± 0.1 | 0.01 ± 0.01 | 44.2% ±0.5 | 57.8% ±0.6 | 80.6% ±0.6 | 63.4% ±0.2 | 87.5% ±0.3 | 83.5% ±0.4 | 18.1% ±0.2 |
| | | Claude 3.5 Sonnet | 1 | 105 | 602 | 1.4 ± 0.2 | 0.01 ± 0.01 | N/A | 52.7% ±0.4 | 81.4% ±0.4 | 68.7% ±0.2 | 89.1% ±0.2 | 82.5% ±0.7 | 12.3% ±0.4 |
| | | Llama 3.1 405B | 1 | 118 | 631 | 1.5 ± 0.1 | 0.01 ± 0.01 | 44.1% ±0.3 | 40.7% ±0.5 | 64.5% ±0.7 | 66.0% ±0.3 | 88.2% ±0.2 | 85.0% ±0.5 | 20.4% ±0.5 |
| | **LM Systems** | MoA | 19 | 25,109 | 17,422 | 15.3 ± 0.3 | 0.06 ± 0.01 | 51.6% ±0.6 | 65.0% ±0.3 | 85.3% ±0.3 | 62.3% ±0.4 | 86.9% ±0.2 | 82.9% ±0.6 | 15.1% ±0.5 |
| | | ADAS | 52 | 72,804 | 44,872 | 58.8 ± 0.3 | 0.63 ± 0.04 | 66.3% ±0.7 | 60.1% ±0.5 | 85.4% ±0.4 | 64.2% ±0.2 | 87.0% ±0.2 | 86.0% ±0.8 | 23.7% ±0.3 |
| | | AFlow | 48 | 68,596 | 41,748 | 55.2 ± 0.4 | 0.59 ± 0.05 | 62.4% ±0.2 | 57.8% ±0.6 | 83.2% ±0.6 | 63.5% ±0.3 | 87.2% ±0.4 | 84.5% ±0.2 | 21.1% ±0.6 |
| | | o1 | Unk. | 112 | Unk. | Unk. | 0.52 ±0.05 | 56.3% ±0.5 | 59.3% ±0.5 | 81.7% ±0.3 | 72.0% ±0.4 | 87.5% ±0.2 | 92.7% ±0.5 | 31.5% ±0.8 |
| **Archon** | **Open Source** | General Purpose | 35 | 51,113 | 31,508 | 3.1 ± 0.3 | 0.12 ± 0.02 | 67.2% ±0.4 | 63.3% ±0.6 | 85.6% ±0.5 | 65.3% ±0.3 | 86.2% ±0.2 | 87.5% ±0.6 | 18.2% ±0.4 |
| | | Task Specific | 44 | 63,157 | 39,949 | 3.7 ± 0.3 | 0.15 ± 0.02 | 71.1% ±0.6 | 68.1% ±0.4 | 89.6% ±0.4 | 67.5% ±0.2 | 88.8% ±0.3 | 89.5% ±0.3 | 28.9% ±0.9 |
| | **Closed Source** | General Purpose | 32 | 52,747 | 27,894 | 40.3 ± 0.5 | 0.44 ± 0.04 | 72.7% ±0.3 | 63.9% ±0.7 | 86.2% ±0.7 | 67.5% ±0.4 | 87.2% ±0.2 | 87.9% ±0.7 | 20.2% ±0.6 |
| | | Task Specific | 40 | 59,085 | 37,271 | 48.2 ± 0.4 | 0.49 ± 0.05 | 77.0% ±0.5 | 68.9% ±0.5 | 90.5% ±0.3 | 72.3% ±0.3 | 89.5% ±0.3 | 92.1% ±0.4 | 25.1% ±0.6 |
| | **All Source** | General Purpose | 35 | 50,427 | 30,461 | 27.8 ± 0.4 | 0.32 ± 0.04 | 76.2% ±0.7 | 66.4% ±0.3 | 89.8% ±0.6 | 69.8% ±0.2 | 87.3% ±0.4 | 89.3% ±0.5 | 23.4% ±0.9 |
| | | Task Specific | 39 | 58,250 | 36,114 | 33.7 ± 0.6 | 0.37 ± 0.04 | **79.5% ±0.4** | **69.0% ±0.6** | **92.5% ±0.5** | **72.7% ±0.3** | **89.7% ±0.2** | **93.5% ±0.6** | **41.4% ±0.7** |

Table 1: **ARCHON's Strong Performance with Open Source, Closed Source, and All Source Models**: Consistent outperformance over SOTA LLMs and LM Systems across explored benchmarks. The standard error numbers were calculated from 10 independent evaluation runs. *MATH and CodeContests use a subset of their test sets for evaluation (Section 4.1).

## 4.1. Benchmarks and Models

**Benchmarks**: We evaluate our models with several benchmarks for instruction-following, reasoning, and coding: MT-Bench (Zheng et al., 2023), AlpacaEval 2.0 (Li et al., 2023), Arena Hard Auto (Li et al., 2024b), MixEval (Ni et al., 2024), MixEval-Hard, MATH (Hendrycks et al., 2021), and CodeContests (Li et al., 2022). We provide an overview of each dataset in Table 22. Since we perform automatic architecture search on a randomly sampled 20% subset of each benchmark, we evaluate on the remaining held-out 80% subset of the benchmark (Table 1) (for ARCHON performances on the entire benchmarks, please see Table 27). The delta between the ARCHON performance on the entire benchmark vs. 80% held-out subset is relatively small: only 0.44%, on average, across these datasets with an S.D. of 0.20%. For MATH, we evaluate a random sample of 200 problems from the dataset's test set. For CodeContests, we evaluate on the 140 test set questions that do not include image tags in the problem description.

**Models**: We test the efficacy of the ARCHON framework by creating different ARCHON architectures across three model categories: 8B or less parameter models, 70B or more parameter models, and closed-source model APIs. For our 8B and 70B+ models, we selected the top-10 performing chat models for each parameter range on the Chatbot Arena Leaderboard (Chiang et al., 2024) as of July 2024. For our ARCHON architectures, we explore multiple model types: open-source, closed-source, and *all-source* (i.e. both open-source and closed-source available). For our closed-source model APIs, we include GPT-4o, GPT-4-Turbo, Claude Opus 3.0, Claude Haiku 3.0, and Claude Sonnet 3.5. We list and compare all of the models tested in the ARCHON framework in Table 28

and Table 29. For all the LLMs utilized and every ARCHON component, we set the generation temperature to 0.7. As baselines, we compare ARCHON against both SOTA single-call LLMs (GPT-4o (OpenAI et al., 2024), Claude 3.5 Sonnet (Anthropic, 2024), and Llama 3.1 405B Instruct (AI@Meta, 2024)) as well as SOTA inference-time approaches (OpenAI's o1 (OpenAI, 2024), MoA (Wang et al., 2024a), ADAS (Hu et al., 2024), and AFlow (Zhang et al., 2024)).

**Task-Specific and General-Purpose ARCHON Architectures**: We compare custom ARCHON architectures, specifically configured to a single evaluation dataset ("Task-specific ARCHON Architectures"), and a generalized ARCHON architecture configured to handle all the evaluation datasets ("General-purpose ARCHON Architectures") (Table 1). For our three model selection settings for ARCHON (i.e. open-source, closed-source, and all-source), we utilize automatic architecture search to find targeted ARCHON architectures for each task (7 architectures total) and find a single generalized ARCHON architecture for maximizing performance over all the tasks (Table 1). The benchmarks are concatenated together and shuffled for generalized ARCHON architecture search. Importantly, all the ARCHON architectures utilized in Section 4 are automatically generated by our Bayesian architecture search technique, which searches over the hyperparameter search space for ARCHON as covered in Section 3.3. For examples of targeted and generalized ARCHON architectures, please see Figure 3 and Appendix A.9. For our architectures, we outline the average number of input tokens (i.e. combined total of tokens inputted over the entire architecture) and output tokens (i.e. combined total of tokens outputted over the entire architecture) for each category in Table 1.

### 4.2. ARCHON vs. Closed-Source LLMs and Other Inference-Time Architectures

**Task Performances**: We start by comparing ARCHON architectures to existing SOTA closed-source LLMs and inference-time architectures across a set of instruction-following, reasoning, and coding tasks. Based on our results in Table 1, we find that ARCHON architectures consistently match or surpass existing approaches across all the benchmarks explored. ARCHON architectures with open-source models demonstrate a 11.2% average improvement over SOTA open-source approaches; for its worst performance, our open-source ARCHON architectures are still 3.1% above SOTA open-source approaches on AlpacaEval 2.0. ARCHON architectures with closed-source models achieve SOTA performance across MT Bench, Arena-Hard-Auto, MixEval, and MixEval-Hard, leading to a 15.1% average improvement over closed-source LMs and a 8.4% average improvement over open-source inference-time frameworks (i.e. MoA, ADAS, and AFlow). Compared to o1 and o1-mini, ARCHON's best targeted architectures beat them by 8.1% and 9.7%, on average, on MT Bench, AlpacaEval 2.0, Arena Hard Auto, MixEval, MixEval Hard, MATH, and CodeContests. For approaches that use all models available, both open and closed-source, ARCHON achieves an average 10.9% improvement over existing SOTA single-call LLMs and an average 8.6% improvement over existing inference-time frameworks.

**Compute Efficiency**: Compared to open-source inference-time frameworks (i.e. AFlow, ADAS, MoA), ARCHON is 20.0% more inference call efficient while having higher performances on all benchmarks tested (Table 1). We also find that our best ARCHON architectures use 15.1% less input tokens and 13.5% less output tokens compared to the best alternative open-source inference-time frameworks. When we utilize ARCHON's architecture search technique with different token budgets (Figure 8), we find that the generated ARCHON architectures achieve 12.4% higher performance than alternate baselines when given the same budget. Overall, the generalized all-source ARCHON architecture achieves 6.4% better performance across all the tasks while being 31% more token efficient than the best LM system baselines (Table 1). Furthermore, compared to the generalized all-source ARCHON architecture, the targeted all-source ARCHON architectures use 15.5% and 18.6% more input tokens and output tokens, respectively, but they achieve 8.4% higher accuracies, on average. The targeted architectures are more compute intensive since they can further leverage additional LM operations towards a single set of specific task constraints (Appendix A.9).

**Discovered Architectures**: We include the targeted and generalized ARCHON architectures in Appendix A.9 (Figure 13). The best performing all-source, general-purpose ARCHON architecture starts with a broad initial layer of our 10 best generators before four successive layers of critique and fusion with Qwen2 72B and Claude 3.5 Sonnet, respectively.

|  | GPQA Diamond | MMLU* | MMLU Pro* |
|---|---|---|---|
| All-Source Generalized AFlow | 37.1%±0.2 | 53.0%±0.4 | 43.4%±0.4 |
| Task-Specific AFlow | 52.4%±0.1 | 71.8%±0.5 | 62.9%±0.5 |
| AFlow Performance Preservation | 70.8% | 73.8% | 67.0% |
| All-Source Generalized ADAS | 39.8%±0.3 | 53.5%±0.3 | 44.1%±0.7 |
| Task-Specific ADAS | 54.4%±0.5 | 73.0%±0.4 | 66.0%±0.4 |
| ADAS Performance Preservation | 73.2% | 73.3% | 66.8% |
| All-Source Generalized ARCHON | 56.1%±0.4 | 76.5%±0.3 | 71.0%±0.1 |
| Task-Specific ARCHON | 61.2%±0.5 | 81.5%±0.3 | 75.4%±0.4 |
| Archon Performance Preservation | 91.7% | 93.9% | 94.2% |

Table 2: **Generalized ARCHON Architecture Strong Performance on Out-of-Domain Tasks**: The generalized ARCHON architectures achieved 91 to 95% the performance of the specialized ARCHON architectures on GPQA, MMLU, and MMLU Pro, despite not being trained for these tasks. Standard error calculated from 10 independent evaluation runs. *For MMLU and MMLU Pro, we use a randomly selected 500 query sample of the test set for evaluation.

Each subsequent layer has fewer fuser models (i.e. 8, 6, and 4), leading to a "funneling" effect on the generations before the final output. The best targeted architectures can vary by task. For instruction-following and reasoning tasks, the targeted architectures tend to be multiple layers of critiquing and fusing with a diverse mix of LMs (Figure 14). For math tasks, the targeted architectures tend to consist of an initial broad set of generations before being reduced quickly to a chosen answer (Figure 15). For coding tasks, the targeted architectures tend to focus on multiple iterations of generation, critique, and fusion over a single response before outputting an answer (Figure 16). Besides the ARCHON architectures included in Appendix A.9, we include all the generalized and targeted ARCHON architectures in our supplementary files.

To explore the efficacy of our general purpose ARCHON architectures, we evaluate them on three previously unseen tasks: GPQA (Rein et al., 2024), MMLU (Hendrycks et al., 2021), and MMLU Pro (Wang et al., 2024b). We find that our all-source general purpose ARCHON architecture captures 91 to 94% of the task-specific ARCHON architectures performances on these benchmarks, suggesting that our architectures are more broadly applicable to out-of-domain tasks (Table 2). The generalized ADAS and AFlow architectures only achieve 66% and 74% of their specialized architecture performance, respectively.

### 4.3. ARCHON by Task

**Instruction-Following and Reasoning**: On MT Bench, AlpacaEval 2.0, and Arena-Hard-Auto, open-source ARCHON architectures outperform current open-source baselines by 10.5%, on average, while closed-source ARCHON outperforms current closed-source baselines by 14.6% (Table 1). With ARCHON, multiple models used for Generators and the depth of fusion layers lead

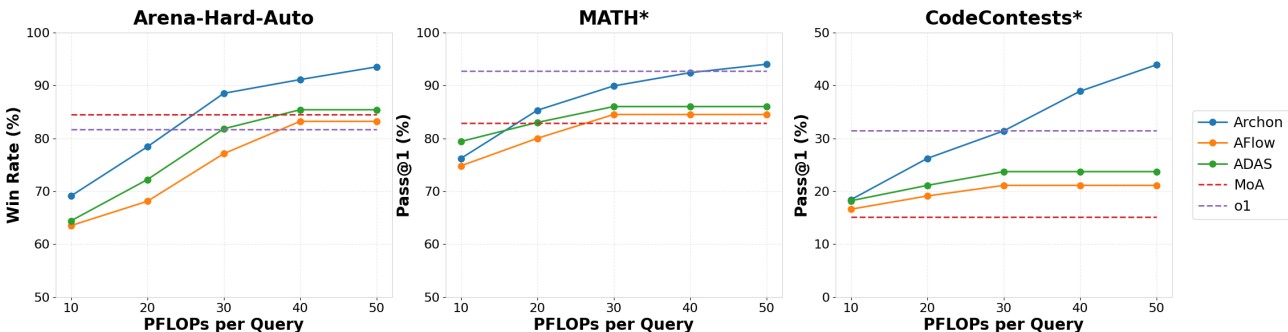

Figure 5: **ARCHON's Performance Exceeds Baselines across FLOP Budgets**: Across different FLOP budgets (Section 3.3), we compare ARCHON architectures against top-performing inference-time system baselines. The MoA architecture and OpenAI's o1 are static so they use the same number of tokens across budgets. The results were averaged over 10 independent evaluation runs. *MATH and CodeContests use a subset of their test sets for evaluation (Section 4.1).

to performance boosts on instruction-following tasks, increasing the richness of responses and allowing multiple iterations for step-by-step instruction-following (Table 30). For reasoning, while the performance boost from ARCHON is smaller when we consider the *aggregate* scores for MixEval and MixEval-Hard, we do see meaningful increases in performance when we create inference-time architectures for each individual task under MixEval and MixEval-Hard (Table 24; Table 25). When we create individual ARCHON architectures for each subtask, we see 3.7 and 8.9 percentage point increases in accuracy, on average, for MixEval and MixEval-Hard, respectively. This finding suggests that reasoning tasks (e.g. math, sciences, logic) require more individualized inference-time architectures.

**Coding**: We have observed that ensembling, fusion, and ranking techniques have limited impact on CodeContests (Figure 4). For example, when we apply the general all-source architecture from Table 22 to CodeContests problems, we achieve small gains from ARCHON (see Table 1). One contributing factor is that, unlike the distribution of instruction-following/reasoning tasks, coding tasks tend to have one or two LLMs that perform substantially better than the rest of models (Table 29). However, when we add unit test generation/evaluation, and scale the number of samples, ARCHON's performance on CodeContests improves significantly (Table 1), allowing us to boost GPT-4o Pass@1 performance by 44.3% for Pass@1 (from 40 to 58 out of 140 questions). For model-based unit test generation/evaluation, we generate 5 unit tests and use the LM to evaluate each candidate response against the generated unit tests, allowing us to rank the different candidate responses (details are provided in Section A.2)

### 4.4. Discussion

**Impact of Model Size**: The ARCHON framework is most effective when utilizing LLMs with 70B+ parameters. When we build ARCHON architectures with 7B open-source mod-

els, we can boost task performance over the best individual 7B LM by 7.5%, on average, compared to the best individual 7B model (Table 32). Across tasks, 7B models work well for ranking but are less effective for critique and fusion.

**Latency and Costs**: Since ARCHON architectures make multiple LLM API calls successively for different operations it can take 5x more time and money than a single LLM API call (Table 33; Table 34). Note that these increases in compute costs and latency translate to higher quality responses, and can be justified in many application domains, such as science, programming, and complex agentic tasks (Rein et al., 2023; Mialon et al., 2023). Furthermore, LLM vendors are rapidly decreasing their inference costs (Table 33). For tasks in which speed is most preferred, future work should explore how distillation strategies (Sreenivas et al., 2024; DeepSeek-AI et al., 2025) could be used to pack the aggregate knowledge of ARCHON architectures into a smaller LM.

## Impact Statement

ARCHON's methodology for optimizing multi-model LM architectures could transform how organizations deploy AI systems, enabling smaller companies and research labs to achieve state-of-the-art performance without access to the largest models or extensive computing resources. The framework's ability to boost performance while reducing compute costs results in energy savings and a lower environmental footprint. The technique of effectively combining multiple specialized models through systematic architecture search could also find applications beyond language models, potentially generalizing to other AI models and tools. As ARCHON demonstrates how to extract significantly better performance from existing models rather than relying on scaling one model alone, it may influence the trajectory of AI development toward more efficient, specialized inference-time architectures rather than ever-larger models.

## Acknowledgments

We thank Simran Arora, Daniel Biderman, Bradley Brown, Ryan Ehrlich, Sabri Eyuboglu, Jordan Juravsky, Jerry Liu, Avanika Narayan, Benjamin Spector, Alyssa Unell, Benjamin Viggiano, and Michael Zhang for their constructive feedback during the composition of the paper. We would also like to thank our collaborators at the Stanford Artificial Intelligence Laboratory (SAIL) and TogetherAI.

We gratefully acknowledge the support of NIH under No. U54EB020405 (Mobilize), NSF under Nos. CCF2247015 (Hardware-Aware), CCF1763315 (Beyond Sparsity), CCF1563078 (Volume to Velocity), and 1937301 (RTML); US DEVCOM ARL under Nos. W911NF-23-2-0184 (Long-context) and W911NF-21-2-0251 (Interactive Human-AI Teaming); ONR under Nos. N000142312633 (Deep Signal Processing); Stanford HAI under No. 247183; NXP, Xilinx, LETI-CEA, Intel, IBM, Microsoft, NEC, Toshiba, TSMC, ARM, Hitachi, BASF, Accenture, Ericsson, Qualcomm, Analog Devices, Google Cloud, Salesforce, Total, the HAI-GCP Cloud Credits for Research program, the Stanford Data Science Initiative (SDSI), and members of the Stanford DAWN project: Meta, Google, and VMWare. The U.S. Government is authorized to reproduce and distribute reprints for Governmental purposes notwithstanding any copyright notation thereon. Any opinions, findings, and conclusions or recommendations expressed in this material are those of the authors and do not necessarily reflect the views, policies, or endorsements, either expressed or implied, of NIH, ONR, or the U.S. Government.

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

# A. Appendix

## A.1. Table of Contents

## A.2. ARCHON LLM Components

| Inference-Time Technique | Definition | Input | Output | Inference Cost | Domains |
|---|---|---|---|---|---|
| Generator | Generates a candidate response from an instruction prompt | Instruction Prompt | Candidate Response(s) | 1 call per cand. | All Domains |
| Fuser | Merges multiple candidate responses into a single response | Instruction Prompt + Candidate Response(s) | Fused Candidate Response(s) | 1 call per cand. | All Domains |
| Critic | Generates strengths/weaknesses for each candidate response | Instruction Prompt + Candidate Response(s) | Candidate Response(s) Strengths/Weaknesses | 1 call | All Domains |
| Ranker | Returns top-K candidate responses | Instruction Prompt + Candidate Response(s) | Ranked Candidate Response(s) | 1 call | All Domains |
| Verifier | Returns the candidate responses with verified reasoning | Instruction Prompt + Candidate Response(s) | Verified Candidate Response(s) | 2 calls per cand. | Reasoning Tasks |
| Unit Test Generator | Generates unit tests to evaluate the candidate responses | Instruction Prompt | Instruction Prompt + Unit Tests | 1 call | Reasoning Tasks |
| Unit Test Evaluator | Uses generated unit tests to evaluate candidate response | Instruction Prompt + Unit Tests + Candidate Response(s) | Scored Candidate Response(s) | 1 call per cand. | Reasoning Tasks |

Table 3: **Overview of ARCHON's Inference-time Techniques**: Definitions, Inputs, Outputs, Costs, and Application Domains.

| Module | Initial Layer Placement | Placement after Initial Layer | >1 Module in Layer | Increase Candidate Responses | Decrease Candidate Responses |
|---|---|---|---|---|---|
| Generator | Yes | No | Yes | Yes | No |
| Fuser | No | Yes | Yes | Yes | Yes |
| Ranker | No | Yes | No | No | Yes |
| Critic | No | Yes | No | No | No |
| Verifier | No | Yes | No | No | Yes |
| Unit Test Generator | No | Yes | No | No | No |
| Unit Test Evaluator | No | Yes | No | No | No |

Table 4: **Rules of ARCHON Construction**: Allowed combinations of each LLM component from Section 3.1.

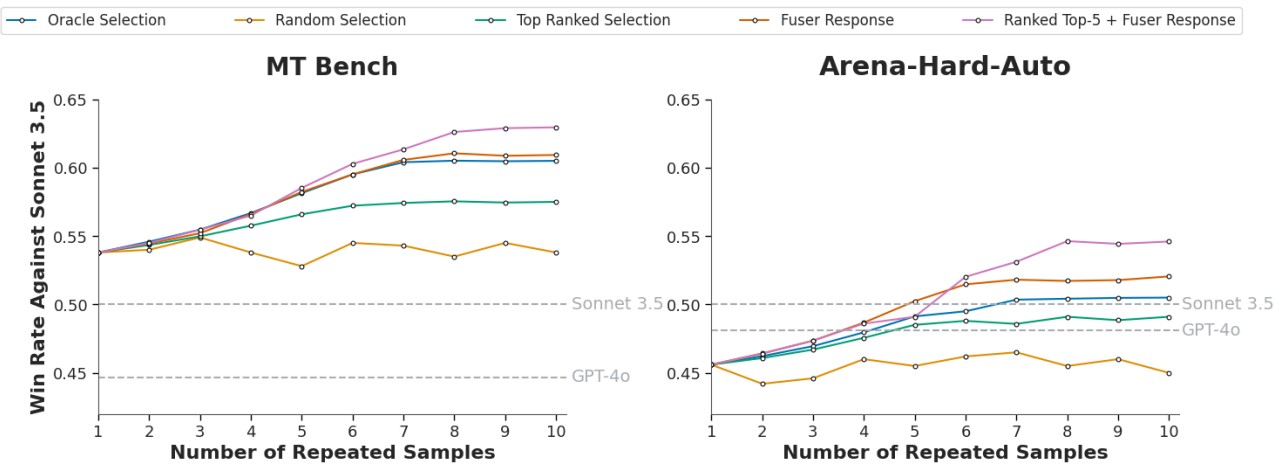

Figure 6: **Performance Gains from Applying Inference Time Techniques on a Single Model**: We repeatedly sample more responses for each individual query. For each sample count, we choose the best response in 5 different ways: **(1)** using an oracle (to get the upper bound for performance of best sample), **(2)** randomly, **(3)** using a ranker model, **(4)** by fusion, in which a model synthesizes a response based on all the samples, and **(5)** by ranking the top-5 best answers and then fusing them. For both MT Bench and Arena-Hard-Auto, we find that fusion is an effective technique. In particular, ranking the candidates first, and then selecting the top-5 and fusing them scores the highest. The best open-source model for these tasks across all the 70B+ models we are considering is WizardLM-2-8x22B (Xu et al., 2024) (see Table 29 for details). For both ranking and fusion, we use Qwen2 72B Instruct (Qwen, 2024).

### A.3. Utilities and Interactions of LLM Components

In this subsection, we present our analysis of the effectiveness of each LLM component (i.e. the *Utility*) and the relationships between each component (i.e. the *Component Interactions*) by evaluating on *instruction-following tasks* (MT Bench, AlpacaEval 2.0, Arena-Hard-Auto), *reasoning tasks* (MixEval, MixEval-Hard, MATH) and *coding tasks* (CodeContests) (Section 4.1). For our ARCHON models, we utilize a host of 70B+ open-source models (Section 4.1; Table 28).

#### A.3.1. GENERATOR

**Utility**: For our Generator module, we find additional model sampling to significantly boost performance (Figure 6), particularly for coding tasks (Table 1). In settings with a limited inference call budget, additional model samples lead to the largest marginal benefit. We see a similar pattern for model ensembling, where sampling from additional models leads to continual performance increases (assuming the models are ordered from best to worst for the given task) (Figure 7).

#### A.3.2. FUSER

**Utility**: For every benchmark explored, we found that the Fuser module substantially improved performance (Figure 6; Figure 7; Figure 4). For the single-generation 10-model ensemble of 70B+ models, the Fuser module improved downstream accuracy by 5.2 points, on average, compared to the single-generation best model (Figure 7). When combined with the Ranker module for ranking the top-5 candidate responses, the Fuser improved downstream accuracy by 7.3 points and 3.6 points, on average, compared to the single-sample best model and the oracle best candidate response, respectively (Figure 7). Overall, we found that Fuser efficacy increased as more candidate responses were provided, demonstrating that additional candidate generations can continue to bolster inference-time architecture performance when combined with a Fuser.

In previous work like Mixture-of-Agents (MoA) (Wang et al., 2024a), multiple layers of Fusers was found to boost performance on some instruction-following tasks (i.e. MT Bench and Alpaca Eval 2.0). Across all the benchmarks explored, we observed similar benefits in the ARCHON framework when adding multiple layers of Fusers (Figure 4). However, based on our results in Figure 12, the number of Fuser layers needed to improve performance varied by task, with some tasks receiving limited benefits from added layers (1-2 point increase in accuracy for MixEval) while others experienced significant benefits with 3-4 fusion layers and more (2 to 5 point increase in win rate for MT Bench and Alpaca Eval 2.0). We attribute

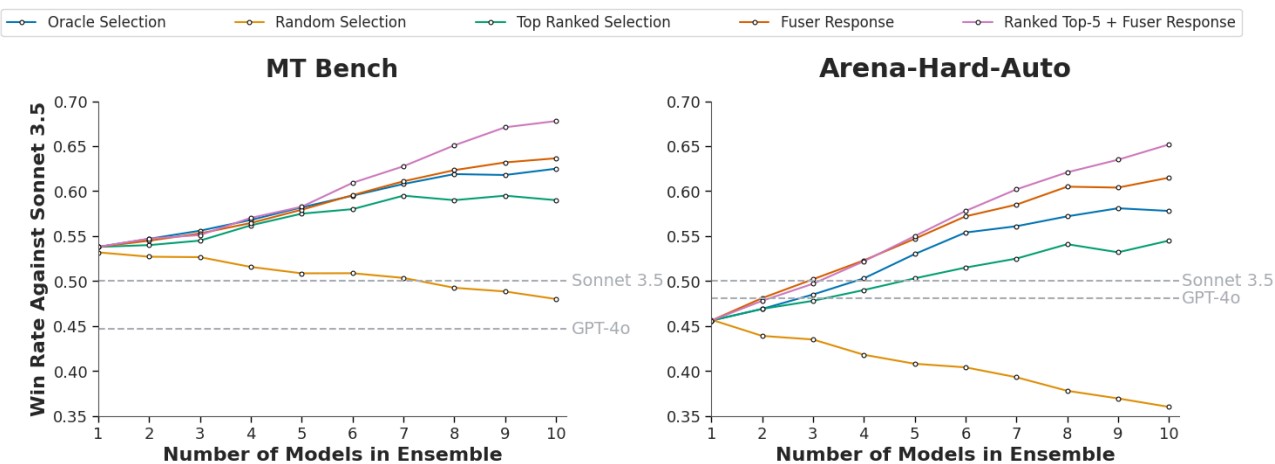

Figure 7: **Performance Gains from Applying Inference-Time Techniques on an Ensemble of Models**: We incrementally add more models to the ensemble, which consists of open-source 70B+ models. The models are added to the pool based on their performance for each task, from best to worse (see Table 29 for details). For each ensemble size, we choose the best response in 5 different modes: **(1)** using an oracle (to get the upper bound for performance of best individual response in the ensemble), **(2)** randomly, **(3)** using a ranker model, **(4)** by fusion, in which one model synthesizes a response based on all the responses of the ensemble models, and **(5)** ranking the top-5 best responses and then fusing them. For MT Bench and Arena-Hard-Auto, we find consistent performance improvements as we add more models to the ensemble. We find that fusion is beneficial across various ensemble sizes and in particular a fused candidate based on the top-5 ranked responses scores highest. The ensemble approach scores higher than applying the same techniques on repeated samples from a single best-performing model (see Figure 6). For both ranking and fusion, we use Qwen2 72B Instruct (Qwen, 2024).

this distinction to the difference in task requirements, with chat and instruction following tasks benefiting more from multiple iterations of revisions through the multiple Fuser layers, leading to greater diversity in the final generation (Table 30).

**Component Interactions**: To better understand how the Fuser module works with the other LLM components, we took the single-sample 10-model ensemble of Generators with a Fuser and tried adding each of these components individually: a Critic, a Ranker, a Verifier, and a Unit Test Generator/Evaluator. Across all of the benchmarks, the added candidate response analyses from the Critic improved the Fuser's ability to effectively merge the different candidate responses, increasing performance by an average of 3.1 percentage points (Figure 4). With the added Ranker, the ARCHON architecture improved the combined Ensemble + Critic + Fuser performance across all the benchmarks by 4.8 percentage points, on average (Figure 4). The Ranker proved most effective for style-oriented tasks (e.g. MT Bench and AlpacaEval 2.0) since the examples mostly focus on improving the instruction-guidance towards the provided prompt. With the added Verifier module (Figure 4), the performance of the Ensemble + Critic + Fuser configuration improved marginally for the instruction-following tasks (1.2 percentage points, on average, for MT Bench, AlpacaEval 2.0, and Arena-Hard-Auto). However, this configuration improved performance more on reasoning tasks (3.2 percentage points for MixEval and MixEval-Hard, on average), assisting generation by filtering out irrelevant or flawed answers before the final fusion step (Figure 4). The added Unit Test Generator and Evaluator was less effective for the instruction-following and reasoning tasks, only providing a 1.5 percentage points increase, on average, when added to the Ensemble + Critic + Fuser configuration (Table 5). However, for coding tasks, we found unit test generation and evaluation significantly improved performance, leading to a 10.7 percentage point increase (56% performance increase comparatively) as we scale model sampling (Table 1).

### A.3.3. CRITIC

**Utility**: The Critic module proved effective for every task we explored in Figure 4 and Table 5. With our 10-model 70B+ Generator ensemble and Fuser configuration of ARCHON, the added Critic improved performance on average by 3.1 percentage points across the benchmarks explored.

**Component Interactions**: While useful for most ARCHON architectures, the added strengths and weaknesses from the Critic module are particularly useful when combined with the Fuser module, helping guide generation fusion for a single layer

| | Model / LLM System | # of Infer. Calls | MT Bench W.R. | AlpacaEval 2.0 L.C. W.R. | AlpacaEval 2.0 Raw W.R. | Arena Hard Auto W.R. | MixEval Hard Acc. | MixEval Acc. | MATH Acc. | Code Contests Acc. |
|---|---|---|---|---|---|---|---|---|---|---|
| Control | Best Open-Source 70B+ Model, Sampled Once | 1 | 55.0% ±0.4 | 44.7% ±0.5 | 37.1% ±0.6 | 45.6% ±0.5 | 58.7% ±0.2 | 86.5% ±0.3 | 84.5% ±0.6 | 22.5% ±0.3 |
| | Ensemble + *Fuser* | 9 | 58.4% ±0.6 | 57.5% ±0.4 | 51.3% ±0.5 | 54.3% ±0.7 | 60.1% ±0.5 | 87.3% ±0.2 | 85.5% ±0.3 | 23.1% ±0.7 |
| | Ensemble + *Critic* + *Fuser* | 10 | 60.9% ±0.3 | 58.7% ±0.6 | 65.8% ±0.3 | 58.8% ±0.4 | 61.7% ±0.5 | 87.4% ±0.3 | 87.2% ±0.5 | 24.9% ±0.4 |
| Ablations | Ensemble + *Ranker* | 9 | 52.5% ±0.7 | 54.7% ±0.5 | 47.6% ±0.4 | 50.5% ±0.6 | 58.7% ±0.3 | 86.8% ±0.4 | 80.4% ±0.4 | 24.1% ±0.4 |
| | Ensemble + *Verifier* | 24 | 53.2% ±0.5 | 56.2% ±0.3 | 50.2% ±0.7 | 52.4% ±0.3 | 55.9% ±0.5 | 85.6% ±0.2 | 85.2% ±0.7 | 25.3% ±0.5 |
| | Ensemble + *Unit Test Gen./Eval.* | 18 | 51.5% ±0.4 | 54.4% ±0.6 | 49.4% ±0.5 | 46.1% ±0.8 | 55.2% ±0.4 | 86.0% ±0.3 | 85.2% ±0.5 | 24.6% ±0.6 |
| | Ensemble + *Ranker* + *Fuser* | 10 | 62.5% ±0.8 | 60.3% ±0.4 | 63.6% ±0.6 | 57.2% ±0.5 | 59.7% ±0.2 | 87.6% ±0.3 | 85.3% ±0.6 | 24.0% ±0.2 |
| | Ensemble + *Verifier* + *Fuser* | 25 | 60.5% ±0.3 | 59.4% ±0.7 | 58.7% ±0.3 | 59.2% ±0.4 | 68.3% ±0.3 | 87.5% ±0.2 | 86.7% ±0.4 | 26.3% ±0.6 |
| | Ensemble + *Unit Test Gen./Eval.* + *Fuser* | 17 | 61.4% ±0.6 | 58.5% ±0.5 | 55.1% ±0.4 | 56.4% ±0.7 | 63.9% ±0.3 | 86.9% ±0.3 | 86.4% ±0.8 | **28.0% ±0.6** |
| | Ensemble + *Critic* + *Verifier* + *Fuser* | 25 | 61.3% ±0.5 | 60.0% ±0.3 | 61.0% ±0.7 | 59.5% ±0.3 | 65.8% ±0.4 | 87.8% ±0.4 | 86.1% ±0.3 | 26.8% ±0.3 |
| | Ensemble + *Critic* + *Ranker* + *Fuser* | 11 | **64.7% ±0.4** | **62.6% ±0.6** | **72.4% ±0.5** | **60.9% ±0.6** | 66.8% ±0.4 | **88.3% ±0.2** | 87.3% ±0.5 | 25.5% ±0.3 |

Table 5: **Impact of Different Compositions of ARCHON's Inference-Time Techniques**: We see increased task performances from adding new LLM components to ARCHON. For CodeContests, we find that there is a single model (Llama 3.1 405B Instruct) that performs considerably better than the rest of the LLMs studied, making it more effective leverage additional model sampling (Table 1). For our ensemble, we use the best 8 open-source 70B+ models for the task (Table 29). For our fuser, critic, ranker, and verifier components, we use the best fuser model found for the task (Table 29). For each evaluation benchmark, we explain its configuration in Table 22 and Section 4.1. The standard error numbers were calculated from 10 independent evaluation runs.

and even useful when placed between multiple fusion layers (on average 3.2 percentage point boost across benchmarks in Figure 4). The Critic module was also effective with the Ranker module, providing additional information for comparing candidate responses (Figure 6) and leading to a 5.9 percentage point increase, on average (Table 5).

### A.3.4. RANKER

**Utility**: From our results in Table 5, Figure 6, and Figure 7, we found the Ranker to be most effective for instruction-following tasks, where pair-wise comparisons of answers focus on style and adherence to the prompt. To examine the candidate selection improvement provided by candidate ranking, we compare three approaches to the Ranker: **(1)** random selection of candidate generation, **(2)** oracle selection of candidate generation, and **(3)** the top-ranked candidate selected by our Ranker. For MT Bench and Arena-Hard-Auto, we find that the ranker improves generation output quality by 3.8% compared to random candidate selection and performs within 2.7% of oracle selection (Figure 6).

**Component Interactions**: Based on our benchmark results in Table 5, the Ranker pairs well with the Critic module; the provided strengths and weaknesses helps guide ranking, particularly for instruction-following tasks, improving performance by 5.9 percentage points, on average. Furthermore, the Ranker was also effective when paired with the Fuser; the filtered list of candidate responses helped improve the final condensed response produced by the Fuser by 3.8 percentage points, on average (Figure 7). When paired with the Verifier and Unit Test Generator, the Ranker had neutral effects; performances changed marginally, either positively or negatively by 1-2 percentage points (Table 5).

Overall, our findings demonstrate the value of added Rankers for instruction-following and reasoning tasks when paired with Fusers. We find that when Rankers are used alone with an ensemble of Generators, their performance lags behind the 10-sample best single model configuration by 3.0 percentage points, on average (Table 5). Additionally, our findings show the importance of building better rankers for more complex reasoning tasks, such as math and coding, which is a challenge also raised by (Brown et al., 2024).

### A.3.5. VERIFIER

**Utility**: The Verifier was most effective for the reasoning benchmarks explored in Table 5. When just using a 70B+ Generator ensemble with Verifier module after generation, the ARCHON configuration lagged behind the ARCHON ensemble and fuser configuration by 1.5 percentage points, on average, across all benchmarks explored. This suggests that the Verifier is most effective when combined with other inference-time techniques.

**Component Interactions**: As noted in Section A.3.2, the Verifier augmented the performance of the Critic and Fuser on reasoning tasks (e.g. Arena-Hard-Auto, MixEval, MixEval-Hard), boosting performance by 3.7 percentage points, on average, when combined together with these modules. Overall, the Verifier is most powerful when augmenting additional components

| | Model / LLM System | # of Infer. Calls | MT Bench W.R. | AlpacaEval 2.0 L.C. W.R. | Arena Hard Auto W.R. | MixEval Hard Acc. | MixEval Acc. | MATH Acc. | Code Contests Acc. |
|---|---|---|---|---|---|---|---|---|---|
| **Control** | Single Generation | 1 | 44.2% ±0.6 | 57.8% ±0.5 | 48.1% ±0.7 | 63.4% ±0.3 | 87.5% ±0.2 | 82.1% ±0.4 | 17.9% ±0.3 |
| | Ensemble + *Fuser* | 11 | 53.7% ±0.3 | 59.5% ±0.6 | 49.7% ±0.5 | 65.5% ±0.2 | 82.0% ±0.3 | 81.0% ±0.6 | 16.0% ±0.4 |
| | Ensemble + *Critic* + *Fuser* | 12 | 56.1% ±0.7 | 59.7% ±0.4 | 53.9% ±0.6 | 67.4% ±0.4 | 82.0% ±0.2 | 82.3% ±0.5 | 18.9% ±0.6 |
| **Ablations** | Ensemble + *Ranker* | 11 | 47.6% ±0.4 | 49.7% ±0.5 | 45.5% ±0.4 | 63.3% ±0.3 | 81.6% ±0.4 | 77.3% ±0.7 | 17.9% ±0.5 |
| | Ensemble + *Verifier* | 11 | 48.4% ±0.5 | 51.2% ±0.7 | 47.7% ±0.8 | 61.4% ±0.2 | 80.5% ±0.3 | 75.5% ±0.3 | 23.0% ±0.4 |
| | Ensemble + *Unit Test Gen./Eval.* | 21 | 46.8% ±0.8 | 49.3% ±0.3 | 41.2% ±0.5 | 60.2% ±0.4 | 80.7% ±0.2 | 78.9% ±0.8 | 24.0% ±0.7 |
| | Ensemble + *Ranker* + *Fuser* | 12 | 58.0% ±0.2 | 60.1% ±0.6 | 52.2% ±0.3 | 65.0% ±0.3 | 82.0% ±0.4 | 82.1% ±0.4 | 18.0% ±0.3 |
| | Ensemble + *Verifier* + *Fuser* | 12 | 55.8% ±0.6 | 54.2% ±0.4 | 60.3% ±0.7 | 67.0% ±0.2 | 82.5% ±0.3 | 83.1% ±0.6 | 22.4% ±0.5 |
| | Ensemble + *Unit Test Gen./Eval.* + *Fuser* | 22 | 56.5% ±0.3 | 61.4% ±0.5 | 51.6% ±0.4 | 67.7% ±0.4 | 81.7% ±0.2 | 84.3% ±0.5 | **25.4% ±0.6** |
| | Ensemble + *Critic* + *Verifier* + *Fuser* | 13 | 56.6% ±0.7 | 62.0% ±0.3 | 55.0% ±0.6 | 68.5% ±0.3 | 82.7% ±0.4 | 85.7% ±0.3 | 22.2% ±0.4 |
| | Ensemble + *Critic* + *Ranker* + *Fuser* | 13 | **60.0% ±0.4** | **62.8% ±0.6** | 56.2% ±0.5 | 69.4% ±0.2 | 88.5% ±0.3 | 87.0% ±0.7 | 18.5% ±0.5 |

Table 6: **ARCHON Component Compositions with GPT-4o**: The ensemble uses generates 10 samples for the given query. The standard error numbers were calculated from 10 independent evaluation runs.

| | Model / LLM System | # of Infer. Calls | MT Bench W.R. | AlpacaEval 2.0 L.C. W.R. | Arena Hard Auto W.R. | MixEval Hard Acc. | MixEval Acc. | MATH Acc. | Code Contests Acc. |
|---|---|---|---|---|---|---|---|---|---|
| **Control** | Single Generation | 1 | 32.1% ±0.7 | 38.5% ±0.5 | 30.4% ±0.6 | 45.2% ±0.3 | 69.5% ±0.2 | 72.3% ±0.5 | 10.5% ±0.6 |
| | Ensemble + *Fuser* | 11 | 44.2% ±0.3 | 43.0% ±0.6 | 40.2% ±0.4 | 46.0% ±0.4 | 73.0% ±0.3 | 70.5% ±0.7 | 6.0% ±0.4 |
| | Ensemble + *Critic* + *Fuser* | 12 | 46.6% ±0.5 | 44.2% ±0.4 | 44.4% ±0.7 | 47.9% ±0.2 | 73.0% ±0.4 | 72.5% ±0.3 | 8.4% ±0.5 |
| **Ablations** | Ensemble + *Ranker* | 11 | 38.1% ±0.6 | 40.2% ±0.7 | 36.0% ±0.5 | 43.8% ±0.3 | 72.1% ±0.2 | 66.2% ±0.6 | 7.5% ±0.4 |
| | Ensemble + *Verifier* | 11 | 38.9% ±0.4 | 41.7% ±0.3 | 38.2% ±0.8 | 41.9% ±0.4 | 71.0% ±0.3 | 68.5% ±0.4 | 19.0% ±0.7 |
| | Ensemble + *Unit Test Gen./Eval.* | 21 | 37.3% ±0.8 | 39.8% ±0.6 | 31.7% ±0.3 | 40.7% ±0.2 | 71.2% ±0.4 | 69.8% ±0.8 | 22.0% ±0.3 |
| | Ensemble + *Ranker* + *Fuser* | 12 | 48.0% ±0.2 | 45.6% ±0.5 | 42.7% ±0.6 | 45.0% ±0.3 | 73.0% ±0.2 | 70.1% ±0.5 | 8.0% ±0.6 |
| | Ensemble + *Verifier* + *Fuser* | 12 | 46.3% ±0.5 | 44.7% ±0.4 | 45.0% ±0.4 | 50.5% ±0.4 | 73.0% ±0.3 | 71.3% ±0.3 | 18.6% ±0.5 |
| | Ensemble + *Unit Test Gen./Eval.* + *Fuser* | 22 | 47.0% ±0.3 | 43.9% ±0.7 | 42.1% ±0.7 | 48.2% ±0.2 | 72.2% ±0.4 | 73.1% ±0.6 | **23.5% ±0.4** |
| | Ensemble + *Critic* + *Verifier* + *Fuser* | 13 | 47.1% ±0.7 | 46.0% ±0.3 | 45.0% ±0.5 | 52.4% ±0.3 | 73.2% ±0.5 | 74.1% ±0.4 | 18.4% ±0.7 |
| | Ensemble + *Critic* + *Ranker* + *Fuser* | 13 | **50.5% ±0.4** | **48.3% ±0.6** | **46.7% ±0.3** | **55.1% ±0.4** | 73.7% ±0.3 | 76.4% ±0.5 | 8.1% ±0.5 |

Table 7: **ARCHON Component Compositions with GPT-4o-mini**: The ensemble uses generates 10 samples for the given query. The standard error numbers were calculated from 10 independent evaluation runs.

| | Model / LLM System | # of Infer. Calls | MT Bench W.R. | AlpacaEval 2.0 L.C. W.R. | Arena Hard Auto W.R. | MixEval Hard Acc. | MixEval Acc. | MATH Acc. | Code Contests Acc. |
|---|---|---|---|---|---|---|---|---|---|
| **Control** | Single Generation | 1 | N/A | 52.7% ±0.4 | 81.4% ±0.6 | 68.7% ±0.3 | 89.1% ±0.2 | 83.5% ±0.5 | 12.5% ±0.3 |
| | Ensemble + *Fuser* | 11 | N/A | 53.0% ±0.6 | 83.2% ±0.4 | 69.5% ±0.2 | 89.0% ±0.3 | 81.8% ±0.6 | 17.0% ±0.4 |
| | Ensemble + *Critic* + *Fuser* | 12 | N/A | 54.2% ±0.3 | 85.4% ±0.7 | 70.9% ±0.4 | 89.5% ±0.2 | 82.6% ±0.4 | 19.4% ±0.6 |
| **Ablations** | Ensemble + *Ranker* | 11 | N/A | 50.2% ±0.5 | 85.7% ±0.5 | 63.8% ±0.3 | 82.1% ±0.4 | 80.2% ±0.7 | 18.5% ±0.5 |
| | Ensemble + *Verifier* | 11 | N/A | 51.7% ±0.7 | 78.2% ±0.3 | 60.9% ±0.2 | 81.0% ±0.3 | 80.1% ±0.3 | 21.0% ±0.4 |
| | Ensemble + *Unit Test Gen./Eval.* | 21 | N/A | 49.8% ±0.4 | 71.7% ±0.8 | 59.0% ±0.2 | 81.2% ±0.2 | 80.9% ±0.8 | 22.0% ±0.7 |
| | Ensemble + *Ranker* + *Fuser* | 12 | N/A | 55.6% ±0.5 | 82.7% ±0.4 | 65.0% ±0.3 | 89.0% ±0.4 | 82.4% ±0.4 | 19.0% ±0.3 |
| | Ensemble + *Verifier* + *Fuser* | 12 | N/A | 54.7% ±0.3 | 85.0% ±0.6 | 70.5% ±0.2 | 89.3% ±0.3 | 84.1% ±0.6 | 21.6% ±0.5 |
| | Ensemble + *Unit Test Gen./Eval.* + *Fuser* | 22 | N/A | 53.9% ±0.6 | 82.1% ±0.5 | 68.2% ±0.4 | 89.2% ±0.2 | 82.0% ±0.5 | **23.5% ±0.6** |
| | Ensemble + *Critic* + *Verifier* + *Fuser* | 13 | N/A | 56.0% ±0.4 | 85.0% ±0.5 | 71.0% ±0.3 | 89.4% ±0.4 | 83.1% ±0.3 | 21.4% ±0.4 |
| | Ensemble + *Critic* + *Ranker* + *Fuser* | 13 | N/A | **58.3% ±0.5** | **86.7% ±0.7** | **73.0% ±0.2** | **89.7% ±0.3** | 85.3% ±0.7 | 19.1% ±0.5 |

Table 8: **ARCHON Component Compositions with Claude 3.5 Sonnet**: The ensemble uses generates 10 samples for the given query. The standard error numbers were calculated from 10 independent evaluation runs.

| | Model / LLM System | # of Infer. Calls | MT Bench W.R. | AlpacaEval 2.0 L.C. W.R. | Arena Hard Auto W.R. | MixEval Hard Acc. | MixEval Acc. | MATH Acc. | Code Contests Acc. |
|---|---|---|---|---|---|---|---|---|---|
| **Control** | Single Generation | 1 | 35.0% ±0.5 | 42.0% ±0.6 | 36.8% ±0.7 | 64.6% ±0.2 | 73.2% ±0.3 | 74.3% ±0.4 | 10.0% ±0.5 |
| | Ensemble + *Fuser* | 11 | 48.2% ±0.3 | 47.0% ±0.4 | 44.2% ±0.5 | 66.5% ±0.3 | 77.0% ±0.2 | 75.1% ±0.7 | 10.8% ±0.3 |
| | Ensemble + *Critic* + *Fuser* | 12 | 50.6% ±0.7 | 48.2% ±0.5 | 48.4% ±0.3 | 68.1% ±0.4 | 77.0% ±0.4 | 76.3% ±0.5 | 11.5% ±0.6 |
| **Ablations** | Ensemble + *Ranker* | 11 | 42.1% ±0.4 | 44.2% ±0.7 | 40.0% ±0.6 | 58.8% ±0.3 | 76.1% ±0.2 | 71.8% ±0.6 | 11.9% ±0.4 |
| | Ensemble + *Verifier* | 11 | 42.9% ±0.6 | 45.7% ±0.3 | 42.2% ±0.8 | 57.9% ±0.2 | 75.0% ±0.3 | 70.5% ±0.4 | 12.0% ±0.7 |
| | Ensemble + *Unit Test Gen./Eval.* | 21 | 41.3% ±0.8 | 43.8% ±0.6 | 35.7% ±0.4 | 55.7% ±0.4 | 75.2% ±0.2 | 74.1% ±0.8 | 13.0% ±0.3 |
| | Ensemble + *Ranker* + *Fuser* | 12 | 52.0% ±0.2 | 49.6% ±0.5 | 46.7% ±0.7 | 60.0% ±0.3 | 77.0% ±0.4 | 75.0% ±0.5 | 12.0% ±0.6 |
| | Ensemble + *Verifier* + *Fuser* | 12 | 50.3% ±0.5 | 48.7% ±0.4 | 48.7% ±0.5 | 67.5% ±0.2 | 77.0% ±0.3 | 77.4% ±0.3 | 10.5% ±0.5 |
| | Ensemble + *Unit Test Gen./Eval. + Fuser* | 22 | 51.0% ±0.3 | 47.9% ±0.7 | 46.1% ±0.6 | 64.2% ±0.4 | 76.2% ±0.2 | 78.3% ±0.6 | **14.3% ±0.4** |
| | Ensemble + *Critic + Verifier + Fuser* | 13 | 51.1% ±0.7 | 50.0% ±0.3 | 49.0% ±0.4 | 68.0% ±0.3 | 77.2% ±0.4 | 77.8% ±0.3 | 10.0% ±0.7 |
| | Ensemble + *Critic + Ranker + Fuser* | 13 | **54.5% ±0.4** | **52.3% ±0.6** | **50.7% ±0.3** | **70.4% ±0.2** | **77.7% ±0.3** | **80.5% ±0.5** | 11.5% ±0.5 |

Table 9: **ARCHON Component Compositions with Claude-3-Haiku**: The ensemble uses generates 10 samples for the given query. The standard error numbers were calculated from 10 independent evaluation runs.

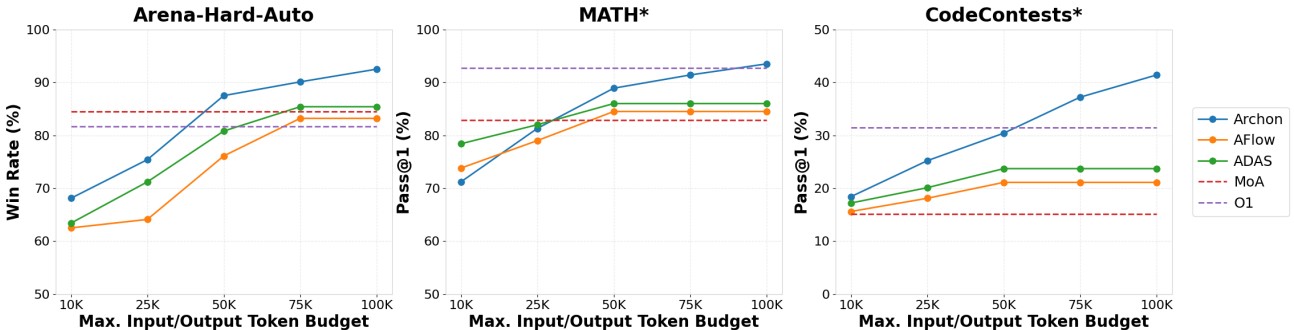

Figure 8: **ARCHON's Performance Exceeds Baselines across Token Budgets**: Across different token budgets (Section 3.3), we compare ARCHON architectures against top-performing inference-time system baselines. The MoA architecture and OpenAI's o1 are static so they use the same number of tokens across budgets. The results were averaged over 10 independent evaluation runs. *MATH and CodeContests use a subset of their test sets for evaluation (Section 4.1).

for tasks requiring verification of intermediate steps and the final response (Table 5). Therefore, the Verifier was less helpful for instruction-following tasks (e.g. MT Bench and AlpacaEval) but more effective for reasoning tasks (e.g. Arena-Hard-Auto and MixEval).

### A.3.6. UNIT TEST GENERATOR AND EVALUATOR

**Utility**: The Unit Test Generator and Evaluator were most effective on reasoning and coding tasks, improving performance on benchmarks that required more verification steps, such as Arena-Hard-Auto, MixEval, MixEval-Hard, MATH, and CodeContests (Table 5). For the reasoning tasks, we found the unit test generator and evaluator to be most effective when combined with other components. When the 70B+ ensemble of Generators was only combined with unit tests, it was less effective for reasoning tasks like Arena-Hard-Auto and MixEval, lagging behind the ensemble and fuser configuration by 3.1 percentage points. This inspired us to look into other inference-time techniques combinations for unit test generation, such as increased sampling and fusion. When we increased generation sampling and added unit test generation/evaluation for CodeContests, we see a 56% boost in Pass@1 performance (Table 1), increasing from 17.9 to 29.3 Pass@1.

**Component Interactions**: When combined with the Fuser module, the Unit Test Generator and Evaluator improved performance by 2.1 percentage points across the benchmarks explored (Table 5). The combined ensemble, Unit Test Generator/Evaluator, and Fuser ARCHON configuration was most effective on the reasoning benchmarks, leading to a 2.5 percentage point boost, on average. For coding, the unit test generator and evaluator was most effective when combined with the best performing Generator (using large sample counts) and a final Fuser (subsection 4.2).

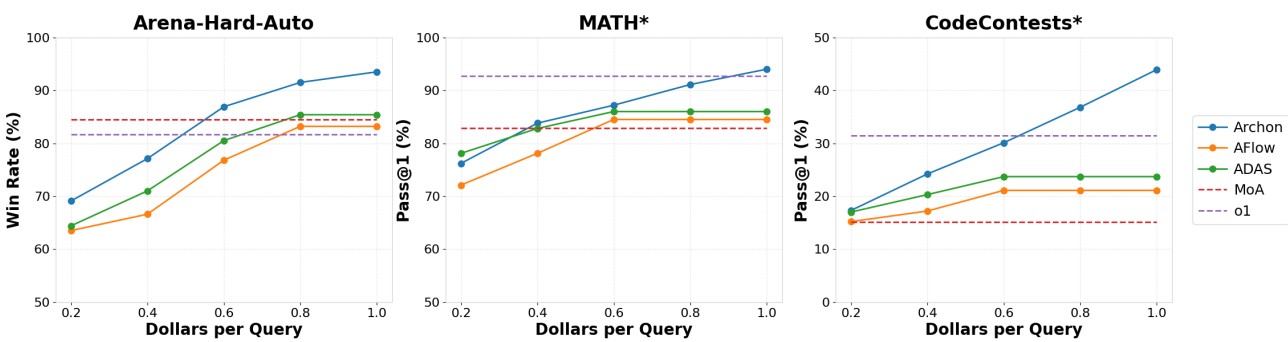

Figure 9: **ARCHON's Performance Exceeds Baselines across Dollar per Query Budgets**: Across different dollar per query budgets (Section 3.3), we compare ARCHON architectures against top-performing inference-time system baselines. The MoA architecture and OpenAI's o1 are static so they use the same number of tokens across budgets. The results were averaged over 10 independent evaluation runs. *MATH and CodeContests use a subset of their test sets for evaluation (Section 4.1).

```
<instruction here>.
```

Table 10: **Generator Prompt**

```
You have been provided with a set of responses with their individual critiques of strengths/weaknesses from various open-source models to the latest user query.
Your task is to synthesize these responses into a single, high-quality response. It is crucial to critically evaluate the information provided in these responses
and their provided critiques of strengths/weaknesses, recognizing that some of it may be biased or incorrect. Your response should not simply replicate the
given answers but should offer a refined, accurate, and comprehensive reply to the instruction. Ensure your response is well-structured, coherent, and adheres
to the highest standards of accuracy and reliability.
Responses from models:
1. <response #1>
Critique: <critique #1>
2. <response #2>
Critique: <critique #2>
...
N. <response #N>
Critique: <critique #N>
<instruction here>
```

((a)) With Critiques

```
You have been provided with a set of responses from various open-source models to the latest user query. Your task is to synthesize these responses into a single,
high-quality response. It is crucial to critically evaluate the information provided in these responses, recognizing that some of it may be biased or incorrect.
Your response should not simply replicate the given answers but should offer a refined, accurate, and comprehensive reply to the instruction. Ensure your response
is well-structured, coherent, and adheres to the highest standards of accuracy and reliability.
1. <response #1>
2. <response #2>
...
N. <response #N>
<instruction here>
```

((b)) Without Critiques

Table 11: **Fuser Prompt: Without and With Critiques**

I will provide you with N responses, each indicated by a numerical identifier []. Rank the responses based on their relevance to the instruction: `<instruction here>`.
`[1] <response #1>`
`[2] <response #2>`
`...`
`[N] <response #N>`
Instruction: `<instruction here>`.
Rank the N responses above based on their relevance to the instruction. All the responses should be included and listed using identifiers, in descending order of relevance to the instruction. The output format should be `[] > []`, e.g., `[4] > [2]`. Only respond with the ranking results, do not say any word or explain.

Table 12: **Decoder-Based Ranking Prompt**

You are a helpful assistant. I will provide you with N responses, each indicated by a numerical identifier (e.g., [1], [2], etc.). Rank the responses based on their relevance to the instruction: `<instruction here>`.
`[1] <response #1>`
`[2] <response #2>`
`...`
`[N] <response #N>`
Instruction: `<instruction here>`.
Evaluate the N responses above based on their relevance to the instruction. All the responses should be included and listed using identifiers. For each response, start the critique with the numerical identifier (e.g., [1]) followed by the strengths and weaknesses. You must include both strengths and weaknesses, even if there are more of one than the other. At the end of each response's analysis, include two new lines to separate the critiques. Do not include any preface or text after the critiques. Do not include any references to previous critiques within a critique. Start with the analysis for the first response and end with the analysis for the last response. All of the N responses should be included and evaluated using identifiers. Structure each response's analysis as follows:
`Strengths:`
`– <strength #1>`
`– <strength #2>`
`– <strength #n>`
`Weaknesses:`
`– <weakness #1>`
`– <weakness #2>`
`– <weakness #n>`

Table 13: **Critic Prompt**

I will provide you with a response indicated by the identifier 'Response'. Provide reasoning for why the response accurately and completely addresses the instruction: `<instruction here>`.
Response: `<response>`
Instruction: `<instruction here>`.
Provide the reasoning for the response above based on its relevance, completeness, and accuracy when compared to the instruction. Do not include any preface or text after the reasoning.

Table 14: **Verifier Prompt**

**Instruction Prompt:** Given the following query, generate a set of N unit tests that would evaluate the correctness of responses to this query.
- The unit tests should cover various aspects of the query and ensure comprehensive evaluation.
- Each unit test should be clearly stated and should include the expected outcome.
- The unit tests should be in the form of assertions that can be used to validate the correctness of responses to the query.
- The unit test should be formatted like 'The answer mentions...', 'The answer states...', 'The answer uses...', etc. followed by the expected outcome.
- Solely provide the unit tests for the question below. Do not provide any text before or after the list. Only output the unit tests as a list of strings (e.g., ['unit test #1', 'unit test #2', 'unit test #3']).
Query: `<instruction here>`

((a)) With Unit Test Cap

**Instruction Prompt:** Given the following query, generate a set of unit tests that would evaluate the correctness of responses to this query.
- The unit tests should cover various aspects of the query and ensure comprehensive evaluation.
- Each unit test should be clearly stated and should include the expected outcome.
- The unit tests should be in the form of assertions that can be used to validate the correctness of responses to the query.
- The unit test should be formatted like 'The answer mentions...', 'The answer states...', 'The answer uses...', etc. followed by the expected outcome.
- Solely provide the unit tests for the question below. Do not provide any text before or after the list. Only output the unit tests as a list of strings (e.g., ['unit test #1', 'unit test #2', 'unit test #3']).
Query: `<instruction here>`

((b)) Without Unit Test Cap

Table 15: **Unit Test Generator Prompt: With and Without Unit Test Cap**

---

**Instruction Prompt:** Compose an engaging travel blog post about a recent trip to Hawaii, highlighting cultural experiences and must-see attractions.
1. Unit Test #1: The blog post mentions at least two cultural experiences specific to Hawaii.
2. Unit Test #2: The blog post highlights at least three must-see attractions in Hawaii.
3. Unit Test #3: The tone of the blog post is engaging and uses descriptive language that would appeal to readers interested in travel.
4. Unit Test #4: The blog post includes factual information about Hawaii's culture, such as local customs, festivals, or historical facts.
5. Unit Test #5: The blog post contains a clear narrative structure, including an introduction, main body, and a conclusion.

---

((a)) Instruction-Following Query

---

**Instruction Prompt:** Alice and Bob have two dice. They roll the dice together, note the sum of the two values shown, and repeat. For Alice to win, two consecutive turns (meaning, two consecutive sums) need to result in 7. For Bob to win, he needs to see an eight followed by a seven. Who do we expect to win this game?
1. Unit Test #1: The response correctly identifies the winning condition for Alice (two consecutive sums of 7).
2. Unit Test #2: The response correctly identifies the winning condition for Bob (a sum of 8 followed by a sum of 7).
3. Unit Test #3: The response explains the probability of achieving two consecutive 7s when rolling two dice.
4. Unit Test #4: The response explains the probability of achieving an 8 followed by a 7 when rolling two dice.
5. Unit Test #5: The response provides a conclusion on who is more likely to win based on the probability analysis.

---

((b)) Reasoning Query

Table 16: **Unit Test Examples**

---

```
Given     the     following     query,     candidate     response,     and     unit     tests,     evaluate     whether     or     not     the     response     passes     each     unit
test.
-  In     your     evaluation,     you     should     consider     how     the     response     aligns     with     the     unit     tests,     retrieved     documents,     and
query.
- Provide reasoning before you return your evaluation.
- At the end of your evaluation, you must finish with a list of verdicts corresponding to each unit test.
- You must include a verdict with one of these formatted options: '[Passed]' or '[Failed]'.
- Here is an example of the output format:
Unit Test #1: [Passed]
Unit Test #2: [Failed]
Unit Test #3: [Passed]
- Each verdict should be on a new line and correspond to the unit test in the same position.
- Here is the query, response, and unit tests for your evaluation:

Query: <instruction here>.

Candidate Response: <response>

Unit Tests:
Unit Test #1: <Unit Test #1>
Unit Test #2: <Unit Test #2>
...
Unit Test #N: <Unit Test #N>
```

---

Table 17: **Unit Test Evaluator Prompt**

## A.4. Bayesian Optimization for ARCHON

### A.4.1. ARCHON SEARCH SPACE AND OBJECTIVE

The ARCHON configuration space can be defined as $\mathcal{X} = \{x_g, x_s, x_f, x_r, x_c, x_v\}$ where:

- $x_g \in [1,10]$ : Number of generator models
- $x_s \in [1,5]$ : Samples per generator (extends to $[1,1000]$ for CodeContests)
- $x_f \in [1,4]$ : Number of fusion layers, including the final fusion layer at the end
- $x_r \in [2,10]$ : Number of models per fusion layer, ranging from 2 to 10 increments of 2
- $x_c \in \{0,1\}$ : Whether to use critic and ranker layers before each fuser
- $x_v \in \{0,1\}$ : Whether to use verification layer before final fusion

The total search space initially contains 18,750 configurations ($10 \cdot 5 \cdot 5^{(4-1)} \cdot 3 = 18,750$), reduced to 9,576 after removing invalid configurations where: 1) initial generations exceed fuser context window (24 candidates); and 2) single fuser layer contains multiple fusers ($x_f = 1$ while $x_r \geq 2$).

Let $f(x)$ be the objective function evaluating an ARCHON configuration $x \in \mathcal{X}$, defined as:

$$f(x) = \text{Performance}(x) - \lambda \cdot \text{Cost}(x) \tag{1}$$

where $\text{Performance}(x)$ is the accuracy on a 20% sample of target tasks and $\text{Cost}(x)$ represents inference compute usage.

### A.4.2. OPTIMIZATION PROCESS

We describe the components of the Bayesian optimization search approach for ARCHON.

First, define $\mathcal{H}$ as a history of architectures and their resulting objective values, which we accumulate throughout the optimization process. We use Expected Improvement (EI) as the acquisition function for determining how to select the next architecture configuration to search:

$$\text{EI}(x; \mathcal{H}) = \mathbb{E}[\max(0, f(x) - f(x^+))], \tag{2}$$

where $f(x^+)$ is the best-observed value of our objective so far for $(x^+, f(x^+)) \in \mathcal{H}$. We use a Gaussian Process model as a surrogate model for approximating $f(x)$.

We now describe the Bayesian optimization process. We first initialize the observation history $\mathcal{H} = \emptyset$. For timestep $t = 1, ..., T$, where $T$ is the maximum number of iterations parameter, we do the following:

1. An architecture $x_t$ is selected using the acquisition function, $x_t = \text{argmax}_{x \in \mathcal{X}} \text{EI}(x; \mathcal{H})$.
2. This architecture $x_t$ is evaluated, and we obtain $f(x_t)$.
3. We use $f(x_t)$ to update our acquisition function and the surrogate model using $H \leftarrow H \cup (x_t, f(x_t))$.

The process continues until either:

- A maximum number of iterations is reached, $T$
- Performance convergence: $|f(x_{n+1}) - f(x_n)| < \epsilon$
- Budget exhaustion: $\text{Cost}(x_1, ..., x_n) > B$

For ARCHON's implementation, we initialize with 230-240 random configurations, as this was found to be optimal through empirical testing. Additional samples beyond this point provide diminishing returns and are better allocated to configuration search. For our implementation, we utilize the Bayesian Optimization python package for global optimization with Gaussian processes.

This formulation allows ARCHON to efficiently explore the configuration space, requiring 88.5% fewer evaluations than greedy search and 90.4% fewer than random search, with Bayesian optimization finding the best architectures in 96.0% of iterations. Traditional greedy search methods may perform comparably for limited inference budgets (<20 calls), but Bayesian optimization becomes increasingly effective as the search space and compute budget grow.

### A.5. Bayes Optimization vs. Alternative Approaches

**Search Techniques**: Within the hyperparameter space, we explored three search algorithms for automating the development of inference-time architectures:

1. **Random Search**: Randomly selects a combination of hyperparameters for our ARCHON architecture.
2. **Greedy Search**: Starting with a base ARCHON configuration, marginally changes each hyperparameter and test if it improves performance or not. If it does, incorporate the change. If not, move on to the next hyperparameter.
3. **Bayesian Optimization**: Efficiently selects the most promising hyperparameter configurations for ARCHON by building a probabilistic surrogate model and leveraging an acquisition function for hyperparameter selection (Snoek et al., 2012; Nardi et al., 2019) (Section A.4).

To get our model ranking for the benchmark, we calculate the model ranking by testing each model individually on a 20% sample of each dataset benchmark in the first stage of the search. To get our fusion model ranking for the benchmark, we use the same approach, testing each model's fusion performance with an ensemble of 10 randomly selected models from the available set. From our experiments, we found that the best generator and fusion models could vary widely dataset to dataset, making it beneficial to perform these rankings for new datasets (Table 29). For search, we use the same 20% sample of each dataset that was used for evaluating generation and fusion, allowing us to guide architecture search with improved evaluation speed while getting meaningful development signal.

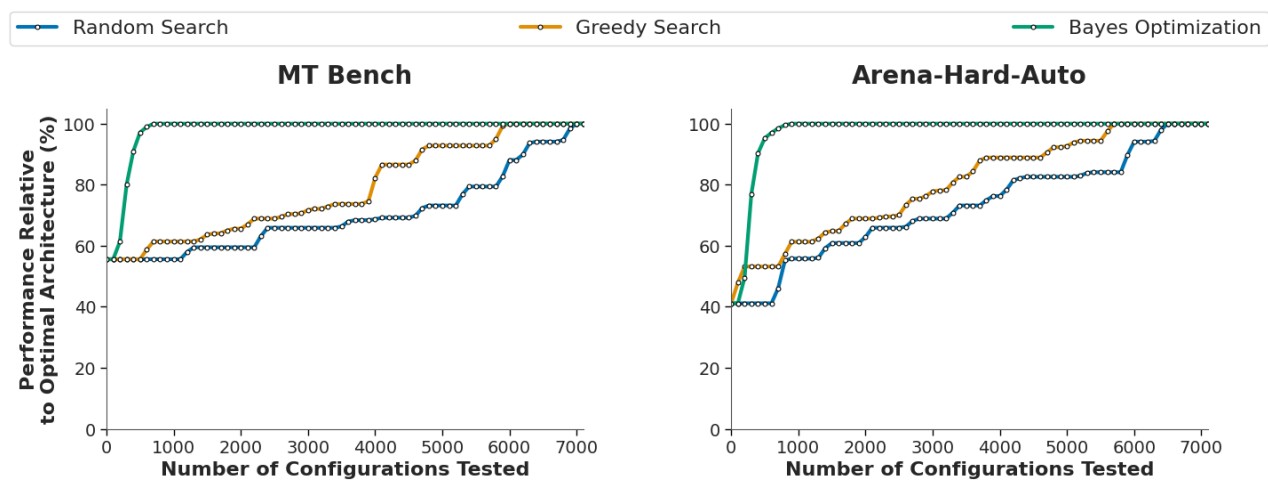

Figure 10: **Impact of Different Optimization Algorithms on ARCHON's Architecture Search**: On the benchmarks MT Bench and Arena-Hard-Auto, we compare four approaches for finding the optimal inference-time architecture: random search, greedy search, and Bayes Optimization. Bayes Optimization finds the optimal architecture in 88.5% less iterations compared to greedy search and 90.4% less iterations compared to random search.

**Comparing Search Algorithms**: In Figure 10, we compare the effectiveness of each search algorithm on our explored benchmarks. While random search guarantees the optimal ARCHON configuration, we found Bayesian optimization to be most effective in terms of tradeoff between finding the optimal configurations and minimizing the number of configurations tested. For 96.0% percent of the search iterations tested in Figure 10, we found that Bayesian optimization had the optimal configuration amongst the four explored search algorithms. We use 230 initial samples for our Bayes Optimization architecture search (Section A.4). Bayesian optimization also found the best architecture configuration in 88.5% less evaluations than greedy search and 90.4% less evaluations than random search.

**Bayesian Optimization Analysis**: In Table 20, we explore how the number of initial testing points, the number of exploration iterations, and the ARCHON inference call budget impacts the effectiveness of Bayesian optimization. Additional initial testing points continue improving search efficacy up until 230-240 samples, where testing would be better delegated towards configuration search. For lower inference call budgets with ARCHON (e.g. <20 inference calls), Bayesian optimization proved less effective, performing more similarly to greedy search or random search given the limited search space (Table 21). Therefore, Bayesian optimization is more effective for more open-ended ARCHON architecture search with larger inference call budgets (e.g. >20 inference calls) whereas traditional component engineering might be better for more limited inference call budgets.

### A.6. ARCHON Architecture Algorithms Comparisons

| # of Init. Points | % of Total Configs | Iter. till Max. Config. | Comb. Iter. |
|---|---|---|---|
| 200 | 2.18% | 353 | 553 |
| 210 | 2.29% | 324 | 534 |
| 220 | 2.40% | 301 | 521 |
| 230 | 2.51% | 284 | 514 |
| 240 | 2.61% | **261** | **501** |
| 250 | 2.72% | 265 | 515 |
| 260 | 2.83% | 256 | 516 |
| 270 | 2.94% | 252 | 522 |

Table 18: MT Bench

| # of Init. Points | % of Total Configs | Iter. till Max. Config. | Comb. Iter. |
|---|---|---|---|
| 200 | 2.18% | 478 | 678 |
| 210 | 2.29% | 431 | 641 |
| 220 | 2.40% | 415 | 635 |
| 230 | 2.51% | **382** | **612** |
| 240 | 2.61% | 389 | 629 |
| 250 | 2.72% | 385 | 635 |
| 260 | 2.83% | 372 | 632 |
| 270 | 2.94% | 368 | 638 |

Table 19: Arena-Hard-Auto

Table 20: **Bayesian Optimization Hyperparameter Comparisons**: On MT Bench and Arena-Hard-Auto, we compare Bayesian optimization configurations for the number of initial sample points. We find that 230 to 240 initial sample points minimizes the combined number of iterations (both initial sampling and exploring) to find the optimal configuration. For the configurations explored, the total number of hyperparameter choices is 9,576.

| | | Iterations to Convergence | | | |
|---|---|---|---|---|---|
| **Inference Budget** | 10 | 20 | 30 | 40 | 50 |
| Random Selection | 387 | 1152 | 2731 | 4359 | 5843 |
| Greedy Search | 343 | 984 | 2153 | 3045 | 4895 |
| Bayes Optimization | **254** | **386** | **452** | **515** | **589** |

Table 21: **ARCHON Architecture Search Algorithms Comparison by Inference Call Budget**: For our comparison, we evaluate on MT Bench.

## A.7. ARCHON Benchmarks and Results

| Benchmark | Example Count | Reference Model | Judge Model | Scoring Type | Metric |
|---|---|---|---|---|---|
| AlpacaEval 2.0 | 805 | GPT-4-Turbo | GPT-4-Turbo | Pairwise Comparison | L.C. & Raw Win Rates |
| Arena-Hard-Auto | 500 | Claude-3.5-Sonnet GPT-4-0314 | GPT-4-Turbo | Pairwise Comparison | Win Rate |
| MT-Bench | 80 | Claude-3.5-Sonnet | GPT-4-0314 | Pairwise Comparison | Adjusted Win Rate |
| MixEval | 2000 | N/A | N/A | Ground Truth | Accuracy |
| MixEval-Hard | 500 | N/A | N/A | Ground Truth | Accuracy |
| MATH | 200 (sampled from 5000) | N/A | N/A | Ground Truth | Pass@1 |
| CodeContests | 140 (non-visual queries) | N/A | N/A | Ground Truth | Pass@1 |

Table 22: **Benchmark Overview**: Evaluation configurations for AlpacaEval 2.0 (Li et al., 2023), Arena-Hard-Auto (Li et al., 2024b), MT-Bench (Zheng et al., 2023), MixEval (Ni et al., 2024), MixEval Hard, MATH (Hendrycks et al., 2021), and CodeContests (Li et al., 2022)

.

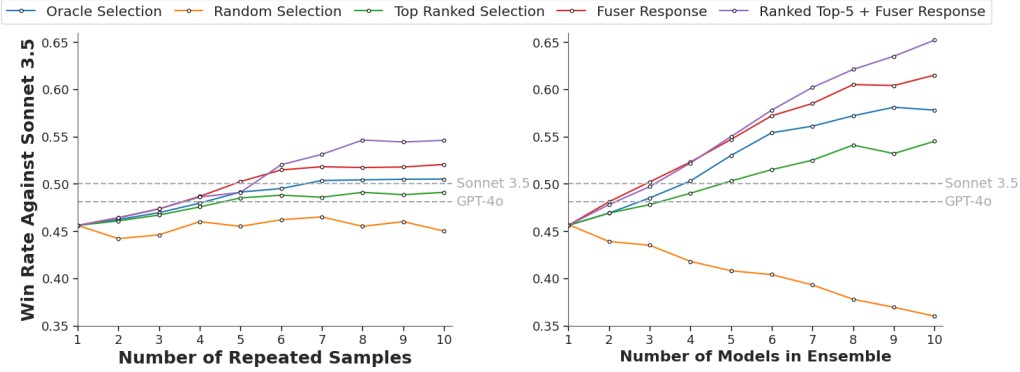

Figure 11: **Performance Gains from Repeated Sampling, Ensembling, Ranking, and Fusing on Arena-Hard-Auto**: The ARCHON win-rate continues to grow significantly as we scale model sampling **(left)** or add additional models to the generator ensemble **(right)**, increasing by 9.3% and 18.5%, respectively. These best results are achieved by selecting the top-5 responses and fusing them. The ensemble models are added based on their individual performance on this task, from best to worse (Table 29). The oracle selection is the performance of picking the best answer generation out of all the generated samples from the ensemble. The results were averaged over 10 independent evaluation runs.

| | Model / LLM System | Arena-Hard-Auto | |
|---|---|---|---|
| | | Score | C.I. |
| | Claude 3.5 Sonnet | N/A | N/A |
| | GPT-4o | 48.1% | (-2.3, 1.8) |
| | Llama 3.1 405B Instruct | 28.4% | (-2.7, 2.5) |
| Open Source | General-purpose ARCHON Architecture | 66.2% | (-2.4, 2.2) |
| | Task-specific ARCHON Architectures | 69.0% | (-2.8, 2.5) |
| Closed Source | General-purpose ARCHON Architecture | 70.5% | (-2.5, 2.0) |
| | Task-specific ARCHON Architectures | 74.4% | (-2.3, 1.6) |
| All Source | General-purpose ARCHON Architecture | 72.5% | (-2.5, 1.8) |
| | Task-specific ARCHON Architectures | **76.1%** | (-1.8, 2.2) |

Table 23: **ARCHON Results on Arena-Hard-Auto Results with Claude-3.5-Sonnet as Baseline Model**: The baseline model is Claude-3.5-Sonnet (default baseline model: GPT-4-0314) while the judge model is GPT-4-Turbo.

| Model / LLM System | Infer. Calls | MixEval - Sub-Datasets | | | | | | |
|---|---|---|---|---|---|---|---|---|
| | | GSM8K | TriviaQA | DROP | MATH | BBH | AGIEval | Average |
| GPT-4o - 2024-05-13 | 1 | 94.9 | 89.1 | 88.2 | **98.5** | 98.3 | 71.5 | 90.3 |
| Claude 3.5 Sonnet | 1 | 98.0 | 92.0 | 92.6 | 96 | 95.6 | 78.0 | 92.0 |
| Llama 3.1 405B Instruct | 1 | **98.2** | 87.9 | 89.6 | 91.5 | 95.8 | 73.2 | 89.6 |
| General-purpose ARCHON Architecture | 29 | 98.3 | 94.8 | 94.6 | 98.1 | 97.3 | 82.1 | 94.2 |
| Task-specific ARCHON Architectures | 34 | **98.2** | **96.7** | **95.6** | **98.5** | **98.8** | **84.2** | **95.7** |

Table 24: **MixEval Results by Sub-Dataset**: For the average computed, we do not introduce any weighting for each dataset.

| Model / LLM System | Infer. Calls | MixEval - Sub-Datasets | | | | | | |
|---|---|---|---|---|---|---|---|---|
| | | GSM8K | TriviaQA | DROP | MATH | BBH | AGIEval | Average |
| GPT-4o - 2024-05-13 | 1 | 72.3 | 70.5 | 70.2 | 94.4 | 80.0 | 53.5 | 73.5 |
| Claude 3.5 Sonnet | 1 | 87.3 | 75.5 | 79.3 | 82.5 | 80.0 | 74.6 | 79.9 |
| Llama 3.1 405B Instruct | 1 | 98.7 | 71.2 | 70.7 | 86.9 | 78.8 | 62.0 | 78.1 |
| General-purpose ARCHON Architecture | 33 | 96.7 | 82.7 | 83.2 | 93.4 | 82.0 | 76.7 | 85.8 |
| Task-specific ARCHON Architectures | 37 | **98.9** | **86.2** | **85.2** | **96.2** | **86.0** | **80.1** | **88.8** |

Table 25: **MixEval-Hard Results by Sub-Dataset**: For the average computed, we do not introduce any weighting for each dataset.

| | GSM8K | MMLU Math | HumanEval Python | MBPP |
|---|---|---|---|---|
| **Model** | Pass@1 | Pass@1 | Pass@1 | Pass@1 |
| GPT-4o | 97.1% | 84.8% | 89.0% | 87.5% |
| Claude 3.5 Sonnet | 96.8% | 90.9% | 90.2% | 88.9% |
| Llama 3.1 405B Instruct | 95.9% | 85.4% | 90.2% | 88.6% |

Table 26: **Additional Math and Code Benchmarks Explored**

| | | | Datasets | | | | | | |
|---|---|---|---|---|---|---|---|---|---|
| | | MT Bench | Alpaca Eval 2.0 | | Arena Hard Auto | Arena Hard Auto | MixEval Hard | MixEval | MATH* |
| **Judge Model** | | GPT-4 0314 | GPT-4 Turbo | | GPT-4 Turbo | GPT-4 Turbo | N/A | N/A | N/A |
| **Reference Model** | | Claude 3.5 Sonnet | GPT-4 Turbo | | Claude 3.5 Sonnet | GPT-4 Turbo | N/A | N/A | N/A |
| **Model / LLM System** | **Infer. Calls** | W.R. | L.C. W.R. | Raw W.R. | W.R. | W.R | Acc. | Acc. | Pass @1 |
| GPT-4o - 2024-05-13 | 1 | 44.7% | 57.5% | 51.3% | 48.1% | 80.3% | 63.6% | 88.0% | 84.5% |
| Claude 3.5 Sonnet | 1 | N/A | 52.4% | 40.6% | N/A | 80.9% | 68.9% | 89.7% | 85.0% |
| Llama 3.1 405B Instruct | 1 | 44.7% | 40.3% | 37.7% | 28.4% | 64.1% | 66.2% | 88.9% | 83.5% |
| MoA | 19 | 51.6% | 65.1% | 59.8% | 52.2% | 84.2% | 62.5% | 87.3% | 82.0% |
| MoA Lite | 7 | 45.6% | 59.3% | 57.0% | 40.6% | 87.8% | 61.1% | 87.1% | 83.0% |
| **Open Source** General-purpose ARCHON Architecture | 35 | 67.5% | 63.0% | 68.3% | 66.2% | 85.1% | 65.5% | 86.9% | 86.5% |
| **Open Source** Task-specific ARCHON Architectures | 44 | 71.6% | 66.7% | 70.7% | 69.0% | 89.5% | 67.5% | 89.6% | 90.5% |
| **Closed Source** General-purpose ARCHON Architecture | 32 | 73.1% | 63.5% | 69.1% | 70.5% | 85.8% | 67.7% | 88.2% | 88.0% |
| **Closed Source** Task-specific ARCHON Architectures | 40 | 77.5% | 68.4% | 72.1% | 74.4% | 90.2% | **72.9%** | 90.4% | 89.5% |
| **All Source** General-purpose ARCHON Architecture | 35 | 76.8% | 65.8% | 70.2% | 72.5% | 89.3% | 70.1% | 88.1% | 90.0% |
| **All Source** Task-specific ARCHON Architectures | 39 | **80.4%** | **67.6%** | **73.3%** | **76.1%** | **92.1%** | **72.9%** | **90.6%** | **93.5%** |

Table 27: **ARCHON's Strong Performance on the Complete Evaluation Datasets after ARCHON Architecture Optimization**: We find that ARCHON's inference-time architectures consistently outperform single-call state-of-the-art LLMs, both open-source and closed-source baselines, when evaluating on the complete benchmarks (Table 22). We explore two configurations: architecture search for building custom ARCHON configurations for each individual benchmark and architecture search for building a single general-purpose ARCHON configuration for all the benchmarks (Section 4.1). We find that a general ARCHON configuration lags behind the custom ones by only 3.2 percentage points, on average, across our all-source settings, which suggests the efficacy of general-purpose inference-time architectures created with our framework. For Arena-Hard-Auto, we also include a configuration with Claude 3.5 Sonnet as a stronger reference model for comparison against ARCHON inference-time architectures and to mitigate bias from GPT judges towards GPT generations. For MT Bench, we use a GPT-4-0314 judge model instead of newer LLM judges to be consistent with previous results on this benchmark. For our task-specific ARCHON architectures, we also provide the average inference calls across the given benchmarks. For our full-list of models explored, please see Table 28. For MATH, we use a randomly sampled subset of size 200 for evaluation (Section 4.1; Table 22). We include our ARCHON architecture results on the held-out 80% subset of each evaluation benchmark in Table 1.

| Models | MT Bench | | Alpaca Eval 2.0 | | Arena Hard Auto | | MixEval | | MixEval Hard | | MATH | | CodeContests | |
|---|---|---|---|---|---|---|---|---|---|---|---|---|---|---|
| | Gen | Fusion | Gen | Fusion | Gen | Fusion | Gen | Fusion | Gen | Fusion | Gen | Fusion | Gen | Fusion |
| GPT-4o | 44.7% | 61.9% | **57.5%** | 64.5% | **48.1%** | 69.2% | 88.0% | **89.4%** | 63.6% | 65.4% | 82.0% | 81.0% | 17.9% | 19.4% |
| GPT-4-Turbo | 42.2% | **63.1%** | 55.0% | **65.8%** | **48.1%** | 61.9% | 88.9% | 89.0% | 64.1% | 64.4% | 79.5% | 73.5% | 9.3% | 14.2% |
| Claude 3 Opus | 30.9% | 57.2% | 40.5% | N/A | 27.0% | 47.9% | 88.3% | 88.2% | 63.6% | 64.0% | 74.5% | 74.0% | 10.0% | 12.5% |
| Claude 3.5 Sonnet | N/A | 71.9% | 52.37% | 63.6% | N/A | **73.2%** | **89.7%** | 89.3% | **68.9%** | **69.5%** | **83.5%** | **86.5%** | 12.1% | 15.5% |
| Qwen 2 72B Instruct | 35.0% | 59.7% | 37.48% | 56.0% | 14.5% | 49.5% | 86.5% | 87.5% | 58.7% | 61.1% | 81.0% | 78.5% | 3.6% | 5.2% |
| DeepSeek LLM 67B Instruct | 18.4% | 20.0% | 17.8% | 17.1% | N/A | N/A | 79.2% | N/A | 42.5% | N/A | 57.0% | N/A | 5.7% | N/A |
| Qwen 1.5 72B Chat | 24.7% | 46.3% | 36.6% | 55.7% | 14.4% | 36.4% | 84.5% | 82.5% | 50.3% | 52.2% | 71.5% | 67.5% | 15.0% | 13.9% |
| Qwen 1.5 110B Chat | 34.4% | 50.3% | 43.6% | 55.9% | 21.9% | 39.7% | 85.3% | 86.5% | 51.8% | 55.6% | 67.0% | 75.5% | 3.6% | 7.8% |
| Wizard 8x22B | **53.8%** | 57.2% | 44.7% | 50.6% | 45.6% | 51.2% | 83% | 78.1% | 54.3% | 50.4% | 76.0% | 60.5% | 7.1% | 10.4% |
| Llama 3.1 8B Instruct | 33.1% | 45.9% | 25.6% | 34.9% | 11.9% | 28.6% | 75.0% | 57.5% | 41.3% | 46.5% | 65.5% | 60.5% | 8.6% | 7.8% |
| Llama 3.1 70B Instruct | 45.0% | 51.9% | 35.6% | 40.2% | 23.8% | 37.2% | 85.7% | 83.5% | 61.1% | 65.5% | 74.0% | 73.5% | 20.7% | **23.4%** |
| Llama 3.1 405B Instruct | 44.7% | N/A | 40.3% | N/A | 28.4% | N/A | 88.9% | N/A | 66.2% | N/A | 78.0% | N/A | **27.1%** | N/A |

Table 29: **ARCHON Generation and Fusion Performances for Single Models**: For Alpaca Eval 2.0, we use the length-controlled win rate (LC WR). For fusion, we gather one candidate from each of the top-10 generator models.

## A.8. ARCHON LLM Analysis

| Model | Source Code | Parameter Count | Max Sequence Length |
|---|---|---|---|
| GPT-4o (OpenAI et al., 2024) | Closed-Source | — | 128K |
| GPT-4-Turbo (OpenAI et al., 2024) | Closed-Source | — | 128K |
| Claude-3-Opus (Anthropic, 2024) | Closed-Source | — | 200K |
| Claude-3.5-Sonnet (Anthropic, 2024) | Closed-Source | — | 200K |
| Claude-3-Haiku (Anthropic, 2024) | Closed-Source | — | 200K |
| Llama-3.1-70B-Instruct (Dubey et al., 2024) | Open-Source | 70B | 8k |
| Llama-3.1-405B-Instruct (Dubey et al., 2024) | Open-Source | 70B | 8k |
| DeepSeek LLM 67B Chat (Guo et al., 2024) | Open-Source | 67B | 32k |
| Qwen2 72B Instruct (Qwen, 2024) | Open-Source | 72B | 32k |
| Qwen1.5 110B Chat (Bai et al., 2023) | Open-Source | 110B | 32k |
| Qwen1.5 72B Chat (Bai et al., 2023) | Open-Source | 72B | 32k |
| Mixtral 8x22B v0.1 (Jiang et al., 2024) | Open-Source | 176B | 32k |
| WizardLM 8x22B (Xu et al., 2024) | Open-Source | 176B | 32k |
| dbrx-instruct (Databricks, 2024) | Open-Source | 132B | 32k |
| princeton-nlp/Llama-3-Instruct-8B-SimPO (Meng et al., 2024) | Open-Source | 8B | 8k |
| princeton-nlp/Llama-3-Instruct-8B-DPO (Meng et al., 2024) | Open-Source | 8B | 8k |
| princeton-nlp/Llama-3-Instruct-8B-RDPO (Meng et al., 2024) | Open-Source | 8B | 8k |
| princeton-nlp/Llama-3-Instruct-8B-IPO (Meng et al., 2024) | Open-Source | 8B | 8k |
| Llama-3.1-8B-Instruct (Dubey et al., 2024) | Open-Source | 8B | 8k |
| Qwen2-7B-Instruct (Qwen, 2024) | Open-Source | 7B | 32k |
| Qwen/Qwen1.5-7B-Chat (Bai et al., 2023) | Open-Source | 7B | 32k |
| mistralai/Mistral-7B-Instruct-v0.2 (Jiang et al., 2023a) | Open-Source | 7B | 32k |
| cognitivecomputations/dolphin-2.2.1-mistral-7b (Hartford, 2024) | Open-Source | 7B | 32k |
| microsoft/Phi-3-mini-4k-instruct (Abdin et al., 2024) | Open-Source | 4B | 4k |
| HuggingFaceH4/zephyr-7b-beta (Tunstall et al., 2023) | Open-Source | 7B | 32k |
| microsoft/Phi-3-small-8k-instruct (Abdin et al., 2024) | Open-Source | 7B | 8k |
| snorkelai/Snorkel-Mistral-PairRM-DPO (Tran et al., 2023) | Open-Source | 7B | 32k |
| mistralai/Mistral-7B-Instruct-v0.3 (Jiang et al., 2023a) | Open-Source | 7B | 32k |

Table 28: **Models Tested with ARCHON**.

| Inference-Time Architecture | Jaccard Similarity (%) | | | | | | |
|---|---|---|---|---|---|---|---|
| | MT Bench | AlpacaEval 2.0 | Arena-Hard Auto | MixEval | MixEval Hard | MATH | Code Contests |
| Best Open-Source 70B+ Model, Sampled 8 Times + Fuser | 45.3% | 52.1% | 48.4% | 55.2% | 58.9% | 65.2% | 63.7% |
| Ensemble (8 Top Models), Sampled Once Each + Fuser | 31.6% | 34.1% | 28.9% | 38.6% | 40.9% | 57.1% | 53.4% |

Table 30: **Jaccard Similarities between Candidates Responses and Fused Response by Benchmark**: For the fuser, we use the best-performing 70B+ model for each benchmark.

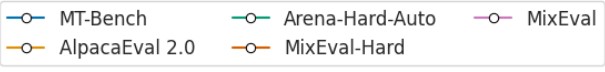

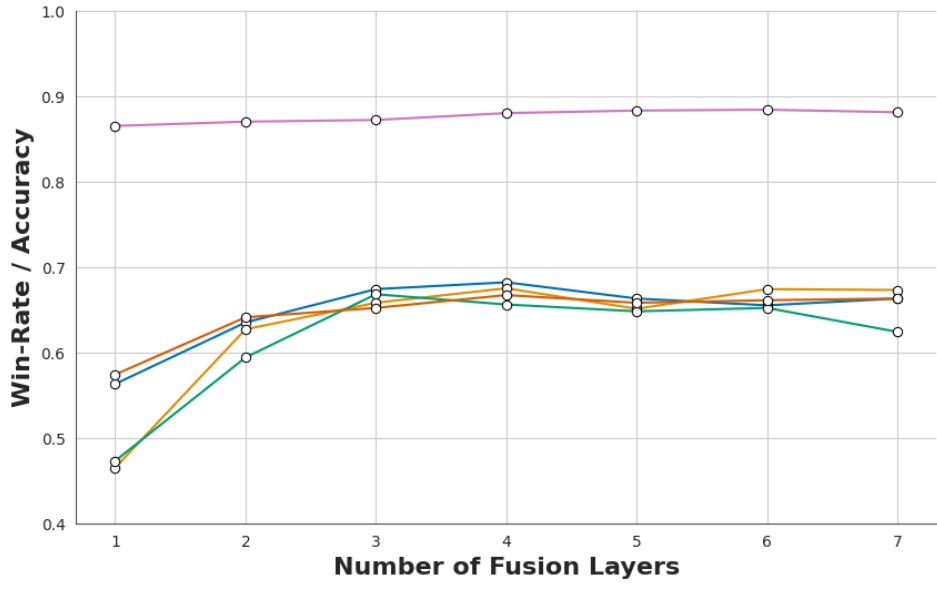

Figure 12: **Fusion Layer Efficacy by Benchmark**: From solely scaling the fusion layers, we see limited benefits across the benchmarks explored but when we add other inference-time techniques, such as Critic and Ranker, we see increased downstream performance as we continue scaling inference-time compute (Figure 4). We use an 8-model ensemble of the top Generator models for each benchmark (Table 29). For our Fuser layers, we use the best Fuser model for the final fuser layer (Table 29). For the intermediate layers, we use the top-8 Fuser models for each benchmark.

## A.9. ARCHON Architectures

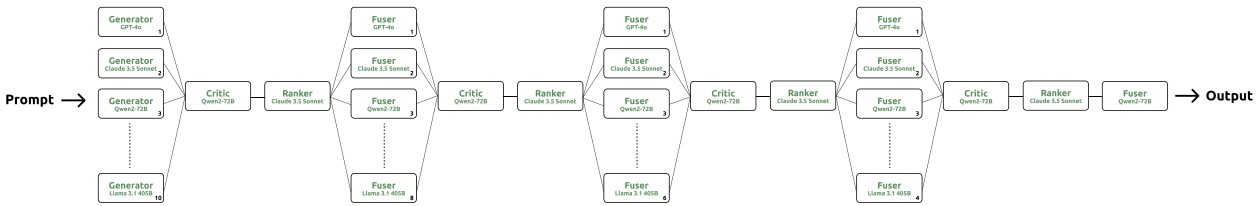

Figure 13: **All-Source Generalizable ARCHON Architecture**: Using ARCHON's architecture search, we found this all-source ARCHON configuration to be effective across the benchmarks explored (except for CodeContests). In the diagram above, we use 10 SOTA all-source LLMs to create multiple successive layers of critic, ranker, and fusers, with each successive fuser layer having less fusers to produce a "funneling" effect as the candidate generations are processed. The layers of critic, ranker, and fuser led to better candidate generations through iterative critique and rewriting. Each of the initial Generator models were sampled once.

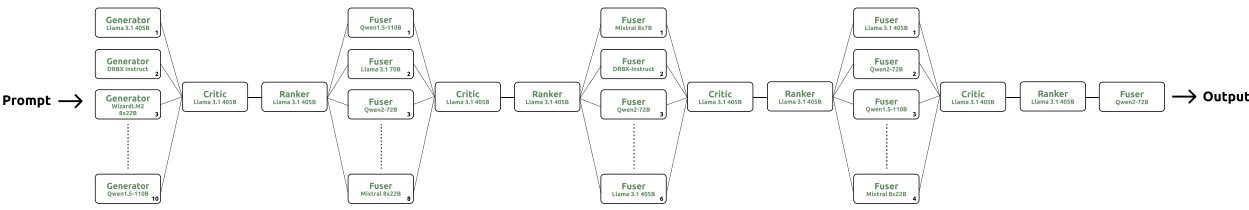

Figure 14: **All-Source ARCHON Architecture for Instruction-Following and Reasoning**: Using ARCHON's architecture search, we found this all-source ARCHON configuration to be effective across the instruction-following benchmarks explored (MT Bench, AlpacaEval 2.0, ArenaHardAuto).

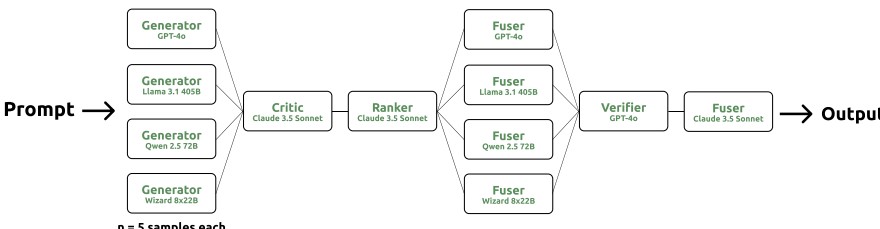

Figure 15: **All-Source ARCHON Architecture for Math**: Using ARCHON's architecture search, we found this all-source ARCHON configuration to be effective across the math benchmarks explored (MATH).

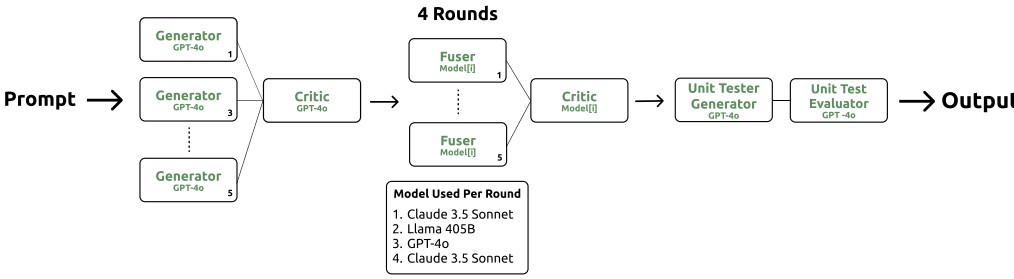

Figure 16: **All-Source ARCHON Architecture for Coding**: Using ARCHON's architecture search, we found this all-source ARCHON configuration to be effective across the coding benchmarks explored (CodeContests).

## A.10. ARCHON by Inference Compute Budget, Model Size, and Cost

| | Number of Inference Calls | Datasets | | | | |
|---|---|---|---|---|---|---|
| | | MT Bench | Alpaca Eval 2.0 | Arena Hard Auto | MixEval | MixEval Hard |
| **70B+ Models** | 1 | 55.0% | 44.7% | 45.6% | 86.5% | 61.1% |
| | 10 | 52.5% | 50.6% | 45.6% | 86.5% | 63.9% |
| | 20 | 65.3% | 60.4% | 59.4% | 89.0% | 65.0% |
| | 30 | 69.2% | 64.5% | **69.0%** | **89.5%** | **67.5%** |
| | 40 | 69.5% | **66.7%** | **69.0%** | **89.5%** | **67.5%** |
| | 50 | **71.6%** | **66.7%** | **69.0%** | **89.5%** | **67.5%** |
| **Closed Models** | 1 | 45.0% | 57.5% | 48.1% | 88.9% | 68.9% |
| | 10 | 57.1% | 63.2% | 68.4% | 90.0% | 70.1% |
| | 20 | 59.4% | 66.5% | 75.5% | **90.6%** | 70.5% |
| | 30 | 70.2% | **68.8%** | **77.4%** | **90.6%** | **72.9%** |
| | 40 | 75.5% | **68.8%** | **77.4%** | **90.6%** | **72.9%** |
| | 50 | **80.4%** | **68.8%** | **77.4%** | **90.6%** | **72.9%** |

Table 31: **ARCHON with Different Inference Budgets**: For AlpacaEval 2.0, we use the length-controlled win rate (LC WR).

| Models / LLM Systems | Datasets | | | | |
|---|---|---|---|---|---|
| | MT Bench | Alpaca Eval 2.0 | Arena Hard Auto | MixEval | MixEval Hard |
| Best 7B Model, 1-Sample | 15.7% | 41.0% | 18.3% | 76.2% | 46.1% |
| Best 7B Model - 10-Sample + Ranking | 16.5% | 43.2% | 18.9% | 78.4% | 48.5% |
| 10-Model, 1-Sample Ensemble + Ranking | **22.4%** | **48.2%** | **25.6%** | **81.5%** | **52.9%** |
| 10-Model, 1-Sample Ensemble + Fusion | 14.3% | 39.4% | 17.5% | 73.2% | 45.2% |
| 10-Model, 1-Sample Ensemble + Top-5 Ranking + Fusion | 15.9% | 41.2% | 18.0% | 75.1% | 46.9% |
| 10-Model, 1-Sample Ensemble + Critic + Fusion | 10.5% | 38.4% | 16.5% | 71.4% | 42.5% |

Table 32: **ARCHON with 7B Open-Source Models**: For AlpacaEval 2.0, we use the length-controlled win rate (LC WR). We use open-source 7B models for testing from Table 28.

| Models | Cost ($) per Million Input Tokens | Cost ($) per Million Output Tokens |
|---|---|---|
| Claude 3.5 Sonnet | $3 | $15 |
| Claude 3.0 Opus | $15 | $75 |
| GPT-4o | $5 | $15 |
| GPT-4-Turbo | $10 | $30 |
| TogetherAI - Llama 3.1 405B Instruct | $5 | $5 |
| TogetherAI - Llama 3.1 70B Instruct | $0.88 | $0.88 |
| TogetherAI - Other Models | $0.90 | $0.90 |

Table 33: **Model API Costs as of November 2024**

| Model / LLM System | Cost ($) per Query for Benchmark | | | | | | |
|---|---|---|---|---|---|---|---|
| | MT Bench | AlpacaEval 2.0 | Arena-Hard Auto | MixEval | MixEval Hard | MATH | Code Contests |
| Claude 3.5 Sonnet | 0.0305 | 0.0171 | 0.0212 | 0.0231 | 0.0226 | 0.0325 | 0.384 |
| GPT-4o | 0.0481 | 0.0236 | 0.0324 | 0.0357 | 0.0361 | 0.514 | 0.562 |
| Llama 3.1 405B Instruct | 0.0281 | 0.0174 | 0.0185 | 0.0212 | 0.0205 | 0.305 | 0.372 |
| General Purpose ARCHON Architecture | 0.364 | 0.189 | 0.195 | 0.284 | 0.252 | 0.375 | 0.461 |
| Task Specific ARCHON Architecture | 0.401 | 0.210 | 0.221 | 0.295 | 0.265 | 0.425 | 0.448 |

Table 34: **ARCHON Costs per Query by Benchmark**

