# OpenReview forum: "An Architecture Search Framework for Inference-Time Techniques"
_ICML.cc/2025/Conference — ICML 2025 poster_

### Official Review · Reviewer_mgTm · 2025-02-25

**Overall Recommendation:** 4

**Summary:**

The paper introduces an inference-time "architecture search" framework (not to be confused with NAS) designed to optimize language model performance on specific benchmarks. The framework systematically selects and integrates multiple inference-time techniques (ensembling, fusion, ranking, critiquing, verification) using Bayesian optimization to find optimal architectures for a given task and compute budget. The framework supports publicly available and closed-source models, showing performance gains over SOTA LLMs and inference-time methods, particularly in instruction-following, reasoning, and coding tasks​.

## update after rebuttal

The authors have answered my concerns with every experiment I found lacking, and even more. Even after reading other reviews, I still believe this paper strongly merits acceptance, as I found most criticisms unconvincing, except for a single one of Uk8K’s, which the authors have responded to in our discussion.
I think the paper has novelty and value. Even if some of the primitives from the paper existed prior to it, an optimized way to use them is very beneficial to practitioners utilizing agents.

**Claims And Evidence:**

All claims are supported as far as I can tell.

**Essential References Not Discussed:**

Not as far as I'm aware.

**Experimental Designs Or Analyses:**

-The experiments use a diverse set of benchmarks (MT-Bench, Arena-Hard-Auto, MixEval, etc.), which adequately cover LLM performance.
Model selection: The comparison against state-of-the-art LLMs (GPT-4o, Claude 3.5 Sonnet, Llama 3.1 405B) ensures a fair baseline.
-The Bayesian optimization search space is well-justified, though further ablation studies could clarify interactions between components​.

**Methods And Evaluation Criteria:**

Evaluation setup: The paper benchmarks Archon against closed-source and publicly available models and existing inference-time techniques (MoA, ADAS, AFlow).
Performance is measured via accuracy and compute efficiency.

**Other Comments Or Suggestions:**

-I appreciate the comprehensive explanation of the Bayesian Optimization (Snoek et al., 2012) algorithm at the appendix.

**Other Strengths And Weaknesses:**

Strengths:
-Automated inference-time architecture search framework integrating multiple techniques provides a way to utilize these various techniques in practice. It is a considerable strength.
-Well written.
-Outperforms other SOTA models and techniques.
-Being able to use both publicly available and closed-source models in the architecture is a plus.

Weaknesses:
-The biggest weakness of the method, in my opinion, is its requirement to get specific benchmarks to optimize the architecture for. While the setting is realistic for certain scenarios where we know exactly what's the type of data we'll encounter, a lot of scenarios require a less specialized approach. With that being said, I'm fine with the paper limiting its scope to the aforementioned subset of scenarios.

-It's unclear to me how the computation budget constraints are enforced during inference and optimized for during the search. From various places across the main body, including A.7.2. it seems the budget consists of model calls. But from various figures (such as Figure 1) it seems the budget is defined as the input/output token ratio (which seems more practical to me, but more challenging to measure). I'd love an explanation, but nonetheless, I think it should be clarified in the paper itself too.

-Minor naming: I'm not sure I'd call Archon an "architecture" as it's a highly overloaded with the model's architecture. Especially since it's not learned. As a reader, I was expecting a NAS paper.

-Minor (not a weakness): there are several "construction rules" that could be changed for cases with higher budgets. e.g., hypothetically speaking, in resource-rich tasks, generators could appear again in later layers to generate later parts in the response after the best candidates were to be chosen from earlier layers (e.g., writing a book chapter by chapter). This comes with a trade-off of expanding the search space.

**Questions For Authors:**

How the computation budget constraints work (see above).

**Relation To Broader Scientific Literature:**

Inference-time architectures are an active area of research, and the paper builds on techniques similar to MoA, RouteLM etc.
Unlike papers that optimize a single model, Archon uses multiple inference-time techniques simultaneously​.

**Theoretical Claims:**

The paper does not introduce new mathematical proofs or theoretical claims.

---

> ### Author Rebuttal · Authors · 2025-04-01
>
> Thank you for taking the time to read our paper! We appreciate your feedback and comments.
>
> We’d like to address each of your concerns individually:
> - *Weaknesses: -The biggest weakness of the method, in my opinion, is its requirement to get specific benchmarks to optimize the architecture for. While the setting is realistic for certain scenarios where we know exactly what's the type of data we'll encounter, a lot of scenarios require a less specialized approach. With that being said, I'm fine with the paper limiting its scope to the aforementioned subset of scenarios.*:
>     - With regards to architecture optimization, we do currently use a dev set of at least 20 queries for optimizing the Archon architecture learned (Section 3.3).
>     - However, the learned architecture can be quite generalizable. For example, by the development sets of the seven benchmarks explored, we were able to create generalized Archon architectures that preserved 90-95% the performance of specialized Archon architectures (Table 1).
>     - This suggests that the learned Archon architectures generalize beyond any individual set of queries and perform well on unseen domains.
>     - To test this hypothesis, we ran an additional set of experiments and evaluated the generalized Archon architectures on three new benchmarks: GPQA, MMLU, and MMLU Pro. These benchmarks cover a variety of topics ranging from science to business to the humanities.
>     - **Note:** We did not use any ground truth labels from these datasets for developing Archon architectures.
>     - We find that our generalized Archon architectures are capable of preserving 91 to 95% the performance gains of specialized Archon architectures trained exclusively for these benchmarks individually.
>     - The generalized ADAS and AFlow architectures only achieve 66% and 73% of their specialized architecture performance, respectively.
>     - We highlight these added results in Table 2 of our revised paper: https://storage.googleapis.com/anonymous-files/archon.pdf.
>
> - *It's unclear to me how the computation budget constraints are enforced during inference and optimized for during the search*:
>     - We impose the constraints by excluding any architecture that would exceed the inference call, input token, or output token budgets from the search space:
>     - Multiple restrictions could be added. For example, you can filter out architectures with more than 20 inference calls or more than 20,000 input tokens.
>     - This prevents our Bayesian optimization algorithm from even considering these invalid architectures in our architecture search (Appendix A.7).
>     - To address your concern, we added a new section to discuss this in Section 3.3 called “Adding Search Restrictions”.
>
> - We agree that the name Archon might elicit concepts surrounding model architectures. However, in our application, we use the term “archon” to highlight that we are using an *architecture of LMs*, rather than an architecture of neurons as in NAS.
> - Also, thank you for highlighting potential avenues for improving the Archon construction rules!
>     - Given an infinite inference budget, future work can further leverage existing techniques, such as generation and fusion, while even developing new components to better translate additional inference compute to improved performance, such as expanded verification systems.
>
> We welcome any further questions. Thank you again for your feedback and comments!  In light of these additional experiments and clarifications to address your comments, we would really appreciate it if you would re-examine our paper and consider raising your score.

---

> > ### Comment · Reviewer_mgTm · 2025-04-02
> >
> > I thank the reviewers for their answers and appreciate their work on my concerns.
> >
> > **I also read other reviewers' feedbacks and still believe this paper merits acceptance**. Most criticisms were unconvincing to warrant rejection, or were replied by the authors adequately in my opinion.
> >
> > The only exception was Uk8K’s following criticsm:
> >
> > >”However, since different LLMs take different costs, it seems inappropriate to unified the budgets as input/output tokens. Subsequently, it is unclear how can we align the budgets of different counterparts for a fair comparison.”
> >
> > I also read the authors’ response.
> > After reading this discussion, I think the best way to measure costs in a unified way is to consider actual $ cost per response. This allows (a) closed-source and open-source models to be directly compared, and; (b) considers each user’s unique costs. For example, I might be able to run a specific model (e.g., Llama-70B) more cheaply due to having a certain better hardware, or because I had a better deal with some compute provider.
> >
> > > To test this hypothesis, we ran an additional set of experiments and evaluated the generalized Archon architectures on three new benchmarks: GPQA, MMLU, and MMLU Pro. These benchmarks cover a variety of topics ranging from science to business to the humanities.
> >
> > I thank the authors for their additional experiment, and agree that it strengthens the claim to the method’s being able to generalize to other benchmarks.
> > To make this point even more convincing, I would try to make additional experiments that make sure the intersection of topics in the “training” set and the “test” set is more limited than this particular experiment.
> >
> > If you fix these two aforementioned points, I believe your paper has more impact than initially believed. Trusting you will, I increase my score.

---

> > > ### Author Response · Authors · 2025-04-03
> > >
> > > **Thank you for taking the time and effort to read our response! We appreciate you raising your score and advocating for our paper’s acceptance.**
> > >
> > > We agree that the average dollar cost per response is a relevant approach for comparing Archon to alternate inference time frameworks and frontier LMs. To address your feedback, we have added a new column to Table #1 and a set of graphs in Figure #9 comparing Archon architectures to ADAS and AFlow architectures by average dollar cost per query across five different budgets, included in our updated PDF: https://storage.googleapis.com/anonymous-files/archon.pdf. In an unrestricted budget setting (Table 1), we find that Archon architectures are 37.1% more cost efficient than ADAS and Archon while achieving 15.1 percentage points better performance across instruction-following, reasoning, math, and coding tasks. In a restricted budget setting (Figure 9), we find that Archon architectures are 44.2% more efficient while achieving 13.4 percentage points better performance across instruction-following, reasoning, math, and coding tasks. For our inference compute providers, we use OpenAI, Anthropic, and TogetherAI.
> > >
> > > To further demonstrate the generalization of Archon architectures, we recompute our scores for MMLU and MMLU Pro after removing their math questions since the Archon architectures were optimized on a different math dataset: the train set of MATH. To do this, we removed questions in the Math category for MMLU Pro and in the Elementary Mathematics, High School Mathematics, College Mathematics, Abstract Algebra, High School Statistics, and Formal Logic categories for MMLU. For GPQA, we do not remove any questions since its graduate level questions in physics, chemistry, biology, and other natural sciences are not related to any datasets used to optimize our Archon architectures.
> > >
> > > Based on our updated results in Table 2, we find that our all-source general purpose Archon architecture captures 91 to 94% of the task-specific Archon architectures performances on these benchmarks, suggesting that our architectures are more broadly applicable to out-of-domain tasks. The generalized ADAS and AFlow architectures only achieve 66 to 74% of their specialized architecture performance, respectively. **Note:** We did not use any ground truth labels from these datasets for developing Archon architectures.
> > >
> > > We welcome any further questions. Thank you again for your feedback and comments!

---

### Official Review · Reviewer_Uk8K · 2025-03-12

**Overall Recommendation:** 2

**Summary:**

This work focuses on how to combine inference-time techniques of LLMs to achieve better performance. It first proposes a framework termed Archon, which is able to incorporate different inference-time techniques rather flexibly. Then, a search method based on Bayesian optimization is designed, which takes as input the target benchmark, the budget for inference, the set of available LLMs, and the available inference-time techniques, and takes as output the layered architecture of different techniques defined in Archon.



### update after rebuttal

I agree with Reviewer mgTm's opinion that "an optimized way to use them is very beneficial", and my concern on the budget alignment is addressed. My remaining concern is not from the technical perspective but lies on the paper organization. IMHO, given that the paper does not introduce new techniques, the motivations and insights that drive this research become rather important, since they may motivate followers and differentiate this manuscript from technical reports. Overall, I am on the boundary of acceptance and rejection, and I would defer to AC's recommendation.

**Claims And Evidence:**

Partially. In the introduction, the authors claimed
> we evaluate the utilities of a comprehensive set of existing and proposed inference-time techniques

> we analyze the interactions between inference-time techniques and explore the benefits of adding new models and new techniques individually.

However, I find that the major analyses/conclusions are not presented in Section 3. In my humble opinion, these are rather important, as they could bring new insights and guide readers to the proposed methods.

**Essential References Not Discussed:**

Sufficient.

**Experimental Designs Or Analyses:**

Overall reasonable. However, since different LLMs take different costs, it seems inappropriate to unified the budgets as input/output tokens. Subsequently, it is unclear how can we align the budgets of different counterparts for a fair comparison.

**Methods And Evaluation Criteria:**

Reasonable.

**Other Comments Or Suggestions:**

The tables and figures are not referred to in order.

**Other Strengths And Weaknesses:**

Overall, this work studies a timely and important topic, the framework is reasonably designed, and the experimental results are positive.

However, my major concern is that the paper heavily relies on the appendix (e.g., Section 3.1 relies on Table 8, Section 3.2 relies on Table 9). This makes the paper hard to follow.

Subsequently, since the analyses are placed in the appendix, there lacks motivation for your architecture design. For example, how such a design achieves optimality? And how does it address the challenges/limitations of previous works?

**Questions For Authors:**

Please elaborate the analyses, which are important for readers to understand the value and motivation of your work.

If this paper is accepted, how can you reorganize the paper so that it is much more readable while fitting the page limit?

**Relation To Broader Scientific Literature:**

Inference-time techniques are important for empowering LLMs.

**Theoretical Claims:**

No theoretical claims.

---

> ### Author Rebuttal · Authors · 2025-04-01
>
> Thank you for taking the time to read our paper! We appreciate your feedback and comments.
>
> *However, I find that the major analyses/conclusions are not presented in Section 3.:*
> - We agree that understanding the utilities of inference-time techniques is central to our contributions. We've enhanced the explanation of ablation findings in our revised draft (highlighted green in Sections 3.1, 3.2): https://storage.googleapis.com/anonymous-files/archon.pdf
> - Our key findings (Table 5, Figures 4, 7, 8) show the following performance increases:
>    - Generation: 10.5% across all tasks when increasing from 1 to 10 generators
>    - Ranking: 5.7% for instruction-following, reasoning, and math tasks
>    - Fusion: 13.2% across all tasks
>    - Critiquing: 8.1% when added before fusion for instruction-following, reasoning, and math
>    - Verification / Unit Testing: 5.4% improvement across all tasks
> - Combining components leads to compounded gains—ranker+critic+fusion yields 15.1% average improvement across all benchmarks and models, while 10+ generators with verification/unit testing boosts math/coding performance by 12.1% (Tables 5-9).
> - These results come from extensive ablation studies across 7 benchmarks and 3 model classes detailed in Appendix A.3 (Tables 18-22).
>
> *For efficiency, it seems inappropriate to use the token budgets:*
> - We used input/output tokens and inference calls for compute-matching Archon because:
>    - True FLOPs of closed-source models used by ADAS, AFlow, and frontier LMs can only be estimated
>    - FLOPs comparisons between different-sized models can be misleading
>    - API providers price by tokens, correlating with their inference FLOP costs
> - Overall, Archon is 20.0% more inference-call efficient, using 15.1% less input tokens and 13.5% less output tokens than ADAS/AFlow while delivering 15.1% better performance (Table #1; Figure #5).
> - Importantly, Archon uses open-source LMs ≤72B parameters, while competitors use GPT-4o with estimated 200B active parameters: https://epoch.ai/gradient-updates/frontier-language-models-have-become-much-smaller
> - To further address your comment, we've added PFLOPs estimates in Table 1 using the estimated active parameters from EpochAI. In unrestricted settings, Archon uses 32.1% fewer FLOPs with models 53% smaller than GPT-4o. In restricted budget settings, Archon achieves equal performance with 38.9% less budget (Figure 6).
>
> *My major concern is that the paper heavily relies on the appendix. If accepted, how can you reorganize the paper for better readability while fitting page limits?:*
> - We appreciate your concerns about appendix reliance. To address your concern, we've:
>    - Added more ablation highlights in the main paper to better explain Archon's design decisions (highlighted in green, Sections 3.2, 3.3)
>    - Added table of contents for easier navigation of Appendix (A.1)
>    - Fixed misordering of Appendix references to corresponding tables/figures
> - These changes make the paper more self-contained while still adhering to final conference page limits.
>
> *How does such a design achieve optimality? How does it address challenges/limitations of previous works?:*
> - The three intellectual contributions underlying Archon are:
>    - Observing that existing inference techniques can be naturally combined, studying their interactions, and optimizing model selection to maximize technique efficacy
>    - Providing an automatic approach for optimal inference technique combinations, yielding significant improvements over SOTA baselines over restricted and unrestricted compute budgets.
>    - Archon architectures generalize to unseen benchmarks, outperforming alternative frameworks like ADAS and AFlow
> - Point (1) motivated Archon's design through ablation studies across 7 benchmarks and 5 model classes (Sections 3.1, 3.2; Appendix A.3). Existing frameworks can't effectively search the inference design space while balancing cost-performance tradeoffs—please note any comparable baselines we missed.
> - Point (2) had never been pursued by previous inference-time papers, allowing us to exceed SOTA LMs and emerging frameworks by +15% accuracy and +30% FLOP efficiency (Table 1; Figures 5-6). While ADAS and AFlow plateaued at 30 PFLOPS per query, Archon continues improving beyond 50 PFLOPs per query.
> - For Point (3), we evaluated generalized Archon architectures on three new out-of-domain benchmarks: GPQA, MMLU, and MMLU Pro. Without using any ground truth labels for architecture adaptation, our generalized architectures preserve 91-95% of the performance gains from specialized architectures trained exclusively for these benchmarks (Table 2). In contrast, generalized ADAS and AFlow architectures only achieve 66% and 73% of their specialized performance.
>
> Thank you again for your feedback and comments! In light of these additional experiments and clarifications to address your comments, we would really appreciate it if you would re-examine our paper and consider raising your score.

---

> > ### Comment · Reviewer_Uk8K · 2025-04-04
> >
> > Thanks for the rebuttal. My concerns are partially addressed. The anonymous revised draft still heavily relies on the appendix. The added texts to Section 3 simply summarize the results in the experiments, yet a self-contained paper should be readable and followable even if we only focus on the main body. Overall, my score is on the borderline.

---

> > > ### Author Response · Authors · 2025-04-04
> > >
> > > Thank you for taking the time and effort to read our rebuttal! We are glad we were able to address your concerns about Archon’s compute-matched comparisons and novelty.
> > >
> > > *The added texts to Section 3 simply summarize the results in the experiments, yet a self-contained paper should be readable and followable even if we only focus on the main body.*
> > >
> > > To address your comment, we now moved all of the experimental findings on individual inference-time component utilities directly to the main body in Section 3. These findings highlight the following trends:
> > >
> > > -**Generator**: Added results showing 10.5% performance gain from increased sampling (1 to 10 models) and its particular effectiveness for coding tasks (56% boost in Pass@1 by scaling from 1 to 1000 generations) (Section 3.1, Figure 7, Table 1)
> > >
> > > -**Fuser**: Included detailed analysis showing 8.9% average improvement across all benchmarks, with evidence that it's most effective when receiving diverse inputs (Section 3.1, Figure 4, Figure 8)
> > >
> > > -**Ranker**: Demonstrated 10.8% improvement for instruction-following and reasoning tasks, with pairwise comparisons focusing on style adherence (Section 3.1, Table 5, Figure 8)
> > >
> > > -**Critic**: Incorporated results showing 11.5% improvement when placed before fusion, with greatest gains on instruction-following benchmarks (Section 3.1, Figure 4, Table 5)
> > >
> > > -**Verifier**: Added performance analysis showing 8.4% improvement across benchmarks, with strongest impact on reasoning tasks (Section 3.1, Table 5)
> > >
> > > -**Unit Test Generator / Evaluator**: Included concrete results showing its critical role in coding tasks (56% boost in Pass@1) (Section 3.1, Table 1, Section 4.2)
> > >
> > > With these added details, Section 3 should be more standalone in its experimental results and analysis while we include the expanded results in the Appendix for future interested readers.
> > >
> > > Thank you again for your feedback and comments! We hope that this addressed your concern about bringing key results back to the paper. Are there any additional comments that we have not addressed? Please let us know if so.

---

### Official Review · Reviewer_TLQP · 2025-03-14

**Overall Recommendation:** 2

**Summary:**

The paper introduces a framework for optimizing inference-time techniques in large language models (LLMs). The contributions, as stated in the paper, are an algorithm that identifies optimal ways to combine inference-time techniques (such as ensembling, ranking, fusion, verification, and critique), as well as understanding the interactions between inference-
time techniques and their utilities.

They evaluate the method on multiple instruction-following, reasoning, and coding benchmarks, demonstrating improvements over state-of-the-art closed-source models.

ARCHON relies on Bayesian optimization to search over a large space of possible architectures, selecting the best-performing configuration for a given task. The results indicate that combining multiple inference-time techniques in a structured manner provides better performance than single techniques alone. The authors release an open-source framework to facilitate further research in this area.

**Claims And Evidence:**

The claim that ARCHON produces architectures that outperform existing state-of-the-art models is well-documented.
The authors report consistent improvements across MT-Bench, AlpacaEval, MixEval, MATH, and CodeContests, often surpassing top closed-source models. However, compute budgets are not always controlled for, making it difficult to isolate whether the improvements come from ARCHON’s methodology or simply from increased inference cost. The improvements shows in Table 1 are quite predictable and not surprising.

Additionally, controlling only for token budget as in Table 1 and Figure 5 doesn't give a clear picture. I understand that, for several models tested, the flops and the throughput are not available, but this causes a lot of issues when comparing the models from an inference budget perspective.

**Essential References Not Discussed:**

-

**Experimental Designs Or Analyses:**

-

**Methods And Evaluation Criteria:**

The evaluation uses multiple diverse benchmarks, covering instruction-following, reasoning, and coding tasks, which strengthens the paper’s claims.
A significant concern as I said is the absence of compute-normalized comparisons. Many baseline models receive far fewer inference calls, and no information is available in terms of flops and generation time (I understand that this is an intrinsic limitation of dealing with closed source models). This does create confounding effects in the experimental results.

**Other Comments Or Suggestions:**

-

**Other Strengths And Weaknesses:**

The open source library can be a very useful contribution for the community and speed up research on test-time compute.

**Questions For Authors:**

-

**Relation To Broader Scientific Literature:**

The contributions are closely related to recent trends in scaling test time compute, although they don't provide significant novelty.

**Theoretical Claims:**

N/A

---

> ### Author Rebuttal · Authors · 2025-04-01
>
> Thank you for taking the time to read our paper! We appreciate your feedback and comments.
>
> *However, compute budgets are not always controlled for... controlling only for token budget as in Table 1 and Figure 5 doesn't give a clear picture:*
> - We appreciate your concerns regarding compute matching experiments. For our comparisons, we originally utilized input/output tokens and inference calls because:
>    - We can only speculate on the true FLOPs of the closed-source models used by baseline approaches (i.e. o1 and GPT-4o for ADAS and AFlow)
>    - Comparing FLOPs between larger and smaller models can be misleading when evaluating efficiency (e.g. 1 PFLOPs with open-source 70B LMs vs. 1 PFLOPs with closed-source 200B LMs)
>    - The API providers, such as OpenAI, Anthropic, Google, Together, etc., price their APIs by input and output tokens per query, correlating directly with their FLOP costs at inference.
> - With our measurements, Archon is 20.0% more inference call efficient, using 15.1% less input tokens and 13.5% less output tokens compared to alternative frameworks (ADAS and AFlow), while delivering 15.1% better performance across all benchmarks (Table #1; Figure #5).
> - Importantly, Archon achieves these gains using open-source LMs of 72B parameters or less, while ADAS and AFlow use GPT-4o with an estimated 200B active parameters per forward pass (https://epoch.ai/gradient-updates/frontier-language-models-have-become-much-smaller).
> - To further address your comment, we've added a column in Table 1 showing PFLOPs per query using estimated active parameters from Epoch AI: https://storage.googleapis.com/anonymous-files/archon.pdf. When benchmarked against ADAS and AFlow in an unrestricted budget setting, Archon architectures use 32.1% less FLOPs while using models that are, on average, 53% smaller than GPT-4o.
> - In a restricted budget setting, this gap increases as we scale the allowed budget, allowing Archon to achieve the same performance as ADAS and AFlow with 38.9% less budget across various settings (Figure 6).
>
> *The improvements shows in Table 1 are quite predictable and not surprising:*
> - We'd like to respectfully push back on this claim. Additional inference compute does not always translate to better task performance (Figure 5). Unlike ADAS and AFlow, which plateau at 30 PFLOPs per query using GPT-4o, Archon continues to improve beyond 50 PFLOPs across all task types (Figure 6): https://storage.googleapis.com/anonymous-files/archon.pdf
> - Even with unlimited budget for inference-time architectures (Table 1, Figures 5-6), Archon architectures outperform alternatives by +15% accuracy while using 32.1% less FLOPs, 20.0% less inference calls, 15.1% less input tokens, and 13.5% less output tokens.
>
> *The contributions are closely related to recent trends in scaling test time compute, although they don't provide significant novelty:*
> - The three intellectual contributions underlying Archon are:
>    - Observing that existing inference techniques can be naturally combined, studying their interactions, and optimizing model selection to maximize technique efficacy
>    - Providing an automatic approach for optimal inference technique combinations, yielding significant improvements over SOTA baselines over restricted and unrestricted compute budgets.
>    - Archon architectures generalize to unseen benchmarks, outperforming alternative frameworks like ADAS and AFlow
> - Point (1) motivated Archon's design through ablation studies across 7 benchmarks and 5 model classes (Sections 3.1, 3.2; Appendix A.3). No existing frameworks effectively search this large inference architecture space while balancing cost-performance tradeoffs—please inform us if we've overlooked comparable baselines.
> - Point (2) had never been pursued by previous inference-time papers, allowing us to exceed SOTA LMs and emerging frameworks by +15% accuracy and +30% FLOP efficiency (Table 1; Figures 5-6). While ADAS and AFlow plateaued at 30 PFLOPS per query, Archon continues improving beyond 50 PFLOPs per query.
> - For Point (3), we evaluated generalized Archon architectures on three new out-of-domain benchmarks: GPQA, MMLU, and MMLU Pro. Without using any ground truth labels for architecture adaptation, our generalized architectures preserve 91-95% of the performance gains from specialized architectures trained exclusively for these benchmarks (Table 2). In contrast, generalized ADAS and AFlow architectures only achieve 66% and 73% of their specialized performance.
>
> We welcome any further questions. Thank you again for your feedback and comments! In light of these additional experiments and clarifications to address your comments, we would really appreciate it if you would re-examine our paper and consider raising your score.

---

### Official Review · Reviewer_Bbe6 · 2025-03-14

**Overall Recommendation:** 2

**Summary:**

This paper proposes a framework called ARCHON for optimizing combinations of inference time techniques to improve the performance of large language models. Their approach combines multiple inference techniques such as ensembles, rankings, etc. and uses automatic architecture fusion search to find the optimal combination of different LLMs.


## update after rebuttal
I appreciate the authors' detailed ablation results and clarifications in the revised version. They have presented more discussion regarding the placement of generation, critique, ranking, fusion, and verification modules, including their "Rules of Construction". This addresses some of my earlier concerns about whether shifting the position of modules can degrade performance. The additional experiments on GPQA, MMLU, and MMLU Pro also help illustrate the robustness of the framework.

However, I still find the overall novelty to be somewhat incremental since ARCHON is largely an engineered combination of known inference techniques. Although the empirical gains are encouraging, the paper would benefit from clearer theoretical or conceptual insight into why specific module orderings are consistently beneficial and how these search strategies might extend to more complex tasks (e.g., advanced math or Olympiad-level problems).

On balance, while the rebuttal clarifies many issues, I remain slightly unconvinced that this submission offers sufficient conceptual novelty for a strong accept. I maintain my recommendation as a weak reject, though I acknowledge the paper’s potential and encourage the authors to strengthen the theoretical framing and expand the demonstration of generality in future submissions.

**Claims And Evidence:**

- The claim that to understand the Utilities of Inference-Time Techniques is not well studied. Lack of the evidence about the improvement percentage from ranking/fusion/verification/etc. with different position. Will shifting the position for different modules causes several performance degradation? Also lack of explanation about the searched framework. How specific the architecture differs from tasks to tasks is not well demonstrated.

**Essential References Not Discussed:**

N/A

**Experimental Designs Or Analyses:**

See concerns under `Claims And Evidence` regarding comprehension score and in-context evaluation.

**Methods And Evaluation Criteria:**

Yes, make sense

**Other Comments Or Suggestions:**

N/A

**Other Strengths And Weaknesses:**

**weakness**
Incremental novelty. This paper is mainly a combination of different strategies. I don't see a strong motivation or insights with explanation or evidence.

**Questions For Authors:**

I'd like to see the framework performance on frontier reasoning datasets such as GPQA, AIME25 and OlympiadBench.

**Relation To Broader Scientific Literature:**

N/A

**Theoretical Claims:**

N.A. in this paper.

---

> ### Author Rebuttal · Authors · 2025-04-01
>
> Thank you for taking the time to read our paper! We appreciate your feedback and comments.
>
> *The claim that Utilities of Inference-Time Techniques is not well studied:*
> - We agree that understanding the utilities of inference-time techniques is central to our contributions. We've enhanced the explanation of ablation findings in our revised draft (highlighted green in Sections 3.1, 3.2): https://storage.googleapis.com/anonymous-files/archon.pdf
> - Our key findings (Table 5, Figures 4, 7, 8) show the following performance increases:
>    - Generation: 10.5% across all tasks when increasing from 1 to 10 generators
>    - Ranking: 5.7% for instruction-following, reasoning, and math tasks
>    - Fusion: 13.2% across all tasks
>    - Critiquing: 8.1% when added before fusion for all tasks
>    - Verification / Unit Testing: 5.4% improvement across all tasks
> - Combining components leads to compounded gains—ranker+critic+fusion yields 15.1% average improvement across all benchmarks and models, while 10+ generators with verification/unit testing boosts math/coding performance by 12.1% (Tables 5-9).
> - These results come from extensive ablation studies across 7 benchmarks and 3 model classes detailed in Appendix A.3 (Tables 18-22).
>
> *Will shifting position for different modules cause performance degradation?:*
> - Our findings in Appendix A.3 (Tables 5-9) show:
>    - Generator components are most effective at architecture start
>    - Critic components work best before ranker/fusion components
>    - Critic, ranker, and fuser components can be stacked sequentially for improvement
>    - Verifier and unit testing components are most effective at architecture end
> - These findings informed our "Rules of Construction" (Section 3.2) to prevent performance-degrading placements. We've highlighted these findings in green in the revised paper and added pointers to the ablation study.
>
> *Lack of explanation about searched framework. How architecture differs across tasks is not well demonstrated:*
> - The all-source generalized Archon architecture is included in Figure 3 (Section 3.2). All generalized and specialized architectures from
> - Table 1 are described in detail in Appendix A.9 (Figures 12-15) and our supplementary code.
> - In Section 4.3 ("Archon by Task"), we discuss task-specific architecture distinctions with these key trends:
>    - Instruction-following tasks benefit from additional generator models and deeper fusion layers (17.8% improvement in win-rate from 1 to 10 generators).
>    - For reasoning, task-specific architectures show meaningful improvements (10.1% improvement for specialized vs. generalized architectures).
>    - For coding, unit testing and increased sample scaling significantly improve performance (44.3% boost in Pass@1).
>    - Instruction-following and reasoning architectures use multiple critique-rank-fuse layers with diverse LMs, while math/coding architectures use repeated samples from a single LM before applying unit-testing or verification.
>
> *Incremental novelty. This paper is mainly combining different strategies without strong motivation or insights:*
> - The three intellectual contributions underlying Archon are:
>    - Observing that existing inference techniques can be naturally combined, studying their interactions, and optimizing model selection to maximize technique efficacy
>    - Providing an automatic approach for optimal inference technique combinations, yielding significant improvements over SOTA baselines over restricted and unrestricted compute budgets.
>    - Archon architectures generalize to unseen benchmarks, outperforming alternative frameworks like ADAS and AFlow
> - Point (1) motivated Archon's design through ablation studies across 7 benchmarks and 5 model classes (Sections 3.1, 3.2; Appendix A.3). No existing frameworks effectively search this large inference architecture space while balancing cost-performance tradeoffs—please inform us if we've overlooked comparable baselines.
> - Point (2) had never been pursued by previous inference-time papers, allowing us to exceed SOTA LMs and emerging frameworks by +15% accuracy and +30% FLOP efficiency (Table 1; Figures 5-6). While ADAS and AFlow plateaued at 30 PFLOPS per query, Archon continues improving beyond 50 PFLOPs per query.
> - For Point (3), we evaluated generalized Archon architectures on three new out-of-domain benchmarks: GPQA, MMLU, and MMLU Pro. Without using any ground truth labels for architecture adaptation, our generalized architectures preserve 91-95% of the performance gains from specialized architectures trained exclusively for these benchmarks (Table 2). In contrast, generalized ADAS and AFlow architectures only achieve 66% and 73% of their specialized performance.
>
> We welcome any further questions! Thank you again for your feedback and comments! In light of the additional experiments and clarifications to address your comments, we would really appreciate it if you would re-examine our paper and consider raising your score.

---

### Decision · Program_Chairs · 2025-05-01

**Decision:**

Accept (poster)

**Comment:**

This paper introduces Archon, a modular framework for inference-time architecture search that composes multiple LLMs and inference-time techniques (e.g., ranking, critique, fusion) to optimize downstream performance under a compute budget. The authors present a thorough experimental evaluation, demonstrating consistent improvements over strong baselines, including GPT-4o, Claude 3.5 Sonnet, and other SOTA systems. While some reviewers expressed reservations regarding the degree of novelty and clarity in the main text, the authors responded comprehensively with expanded ablations, generalization experiments, and additional cost-efficiency analyses. Inference-time scaling techniques abound but application builders are flummoxed as to how to use them and when. Archon is timely and opens up further research along this direction.